# Impaired cerebellar Purkinje cell potentiation generates unstable spatial map orientation and inaccurate navigation

Julie Marie Lefort [1], Jean Vincent[1], Lucille Tallot[1], Frédéric Jarlier[1], Chris Innocentius De Zeeuw[2,3], Laure Rondi-Reig[1,4] & Christelle Rochefort[1,4]

Cerebellar activity supported by PKC-dependent long-term depression in Purkinje cells (PCs) is involved in the stabilization of self-motion based hippocampal representation, but the existence of cerebellar processes underlying integration of allocentric cues remains unclear. Using mutant-mice lacking PP2B in PCs (L7-PP2B mice) we here assess the role of PP2B-dependent PC potentiation in hippocampal representation and spatial navigation. L7-PP2B mice display higher susceptibility to spatial map instability relative to the allocentric cue and impaired allocentric as well as self-motion goal-directed navigation. These results indicate that PP2B-dependent potentiation in PCs contributes to maintain a stable hippocampal representation of a familiar environment in an allocentric reference frame as well as to support optimal trajectory toward a goal during navigation.

[1] Sorbonne Université, UPMC CNRS, INSERM, Neurosciences Paris Seine, NPS, Institut de Biologie Paris Seine, IBPS, Cerebellum Navigation and Memory Team, CeZaMe, F-75005 Paris, France. [2] Netherlands Institute for Neuroscience, Royal Netherlands Academy of Arts and Sciences, 1105 BA Amsterdam, The Netherlands. [3] Department of Neuroscience, Erasmus MC, 3000 CA Rotterdam, The Netherlands. [4]These authors contributed equally: Laure Rondi-Reig, Christelle Rochefort. Correspondence and requests for materials should be addressed to L.R-R. (email: laure.rondi-reig@upmc.fr)

Hippocampal neurons provide a population code for space. Key elements of this representation are hippocampal place cells, the firing rate of which forms place fields at particular location of the environment[1]. An ensemble of place cells can represent a local environment[2] and remap following variations of either the environmental shape or the available sensory cues[3,4]. As recently reviewed by Kelemen and Fenton[5], a switch between multiple hippocampal representations has been observed on different time scales. The existence of dynamics in hippocampal activity between different coordinate systems (i.e., allocentric versus body-centered reference frame) classically referred to remapping was revealed by environmental manipulations or modifications of navigation strategy[6–12]. Spatial map rotation has also been observed without changes in the environment in aged rats[13] or following disorientation[8]. Such events, when occurring in a familiar environment, have been proposed to depend on the anchoring of place cell ensemble on different sets of sensory cues, i.e., local or distal external cues as well as self-motion signals[14–16]. However, the rules by which sensory cues are weighted to anchor the hippocampal representation in a particular coordinate system and to stabilize it during exploration of a familiar place remain unclear.

The cerebellum has recently been shown to play a critical role in monitoring multimodal sensory information via the granule cells-parallel fiber Purkinje cells pathway[17]. Multimodal sensory information is integrated at the level of individual cerebellar granule cells[18,19]. Purkinje cells have been shown to transform head-centered signals originating from the vestibular afferents into earth-referenced signals required for spatial orientation[20,21]. In addition, in the cerebellar nuclei relaying Purkinje cells output, fastigial neurons differentially encode self-generated and externally applied self-motion, a crucial feature to simultaneously control posture and accurately perceive self-motion[22]. Finally, cerebellar activity can impact hippocampal activity through two major pathways (see ref. [17] for review). First, lobules IX-X flocculus and paraflocculus project to vestibular nuclei as well as to the Prepositus Hypoglossi Nucleus, directly feeding the head-direction cell system[23]. Second, the posterior cerebellar lobules (including VI and Crus I) project, through the deep cerebellar nuclei as well as through the ventro and centro-lateral thalamus[24] to the parietal cortex which contains movement cells[25] and path cells[26,27].

Using the L7-PKCI transgenic mouse model specifically deficient for long-term depression plasticity (LTD) at parallel fiber-Purkinje cell synapses, we previously showed an implication of PKC-dependent LTD in processing self-motion signals and in subsequently shaping the hippocampal spatial code. Indeed, L7-PKCI mice present impaired hippocampal place cell properties when relying on self-motion cues[28] and non-optimal trajectories toward a goal[29,30]. Whereas these results clearly demonstrated that the cerebellum interacts with hippocampal functions, they also raised the question of how the computation undertaken at the level of Purkinje cells influences hippocampal place cells activity. In order to address this, we here investigated whether and how diverging forms of cerebellar Purkinje cells computations differentially affect the hippocampal spatial code and navigation. To this end, we used L7-PP2B mice, a transgenic mouse model in which PP2B-dependent processes are abolished[31]. This includes both synaptic and intrinsic forms of Purkinje cells potentiation, and thereby alters cerebellar computations in a markedly different manner from L7-PKCI mice. Taking advantage of this unique feature of L7-PP2B mice, we explored the consequences of such dysfunction on hippocampal spatial representation during free exploration in a circular arena as well as on goal-directed navigation performances in a Water Maze navigation task.

We found that L7-PP2B mice exhibit an unstable hippocampal representation that occurs specifically when mice have to orient their spatial map according to the allocentric cue. At the behavioral level, L7-PP2B mice are impaired in both allocentric and self-motion goal-directed navigation. These results indicate that PP2B-dependent potentiation in cerebellar Purkinje cells contributes to maintain a stable hippocampal representation of a familiar environment in an allocentric reference frame as well as to support optimal trajectory toward a goal during navigation.

## Results

**Experimental design**. Transgenic L7-PP2B mice, specifically knocked-out for calcineurin (PP2B) in Purkinje cells, were obtained by crossing floxed CNB1 mice (regulatory subunit of calcineurin) with a Purkinje cell-specific (L7-) cre-line[31,32]. These mice showed preserved sensori-motor properties, but impaired motor adaptation abilities on a rod rotating at constant or accelerating speed (Supplementary Fig. 1). Neural activity was sampled in 8 L7-PP2B and 8 littermate control male mice during free motion in a circular arena containing a salient cue (a card with a bottle affixed to it). This proximal cue serves as a unique directional reference in two consecutive 12-min sessions. Between sessions, mice were returned to their home cage and the arena was cleaned to homogenize potential odor-cues.

**Unstable hippocampal spatial representation in L7-PP2B mice**. Behavioral analyses and place cell properties were first analyzed on sessions S1. Exploration of the familiar arena was not different between controls and L7-PP2B mice as illustrated by their general locomotor activity (traveled distance, mean speed, and number of stops) or their center versus periphery exploratory behavior (distance from wall, time spent in the center, and number of entries in the center, Supplementary Fig. 2, all $p > 0.05$, Mann–Whitney $U$-test). A total of 412 control and 442 L7-PP2B dorsal CA1 hippocampal pyramidal cells (Supplementary Fig. 3) were recorded among which 211 and 195 place cells were analyzed respectively. Analyses of place cell firing properties revealed a larger field size, an impaired spatial coherence (Supplementary Fig. 4a–c, Mann–Whitney $U$-test, $p < 0.005$ for both parameters) as well as a non-significant decrease of spatial information content in L7-PP2B mice (Supplementary Fig. 4d, $p = 0.066$, Mann–Whitney $U$-test). In contrast, intra-session stability (stability within a session), mean field and peak firing rates were similar to control mice (Supplementary Fig. 4e–g).

Further analyses performed on consecutive S1–S2 sessions revealed a deficit in inter-session stability (spatial correlation between sessions) in L7-PP2B mice (controls, $0.67 \pm 0.02$, L7-PP2B, $0.46 \pm 0.03$, $U = 14604$, $p < 10^{-6}$, Mann–Whitney $U$-test). The bimodal distribution of S1–S2 similarity coefficient suggested the existence of a subpopulation of unstable place fields (Fig. 1b, c).

This distribution was significantly different between control and L7-PP2B mice (Kolmogorov–Smirnov, $p < 0.001$). These low S1–S2 similarity coefficients were associated with random S1–S2 rotation angle, which allowed to identify unstable cells with a K-means algorithm using the two dimension vectors [S1–S2 angle; S1/S2 similarity coefficient]. The proportion of unstable place fields was higher in L7-PP2B mice compared to controls (Fig. 1d, e; 10/211 for controls, 47/195 for mutants, Fisher exact test, $p < 0.0001$). In fact, when they occurred, instabilities between sessions S1 and S2 were observed on ensembles of recorded place cells (Supplementary Fig. 5a–b). Therefore, we further investigated this phenomenon at the level of place cell populations. We found that during a given session in which place cells were simultaneously recorded, rotation angles were similar (Supplementary Fig. 5c–d),

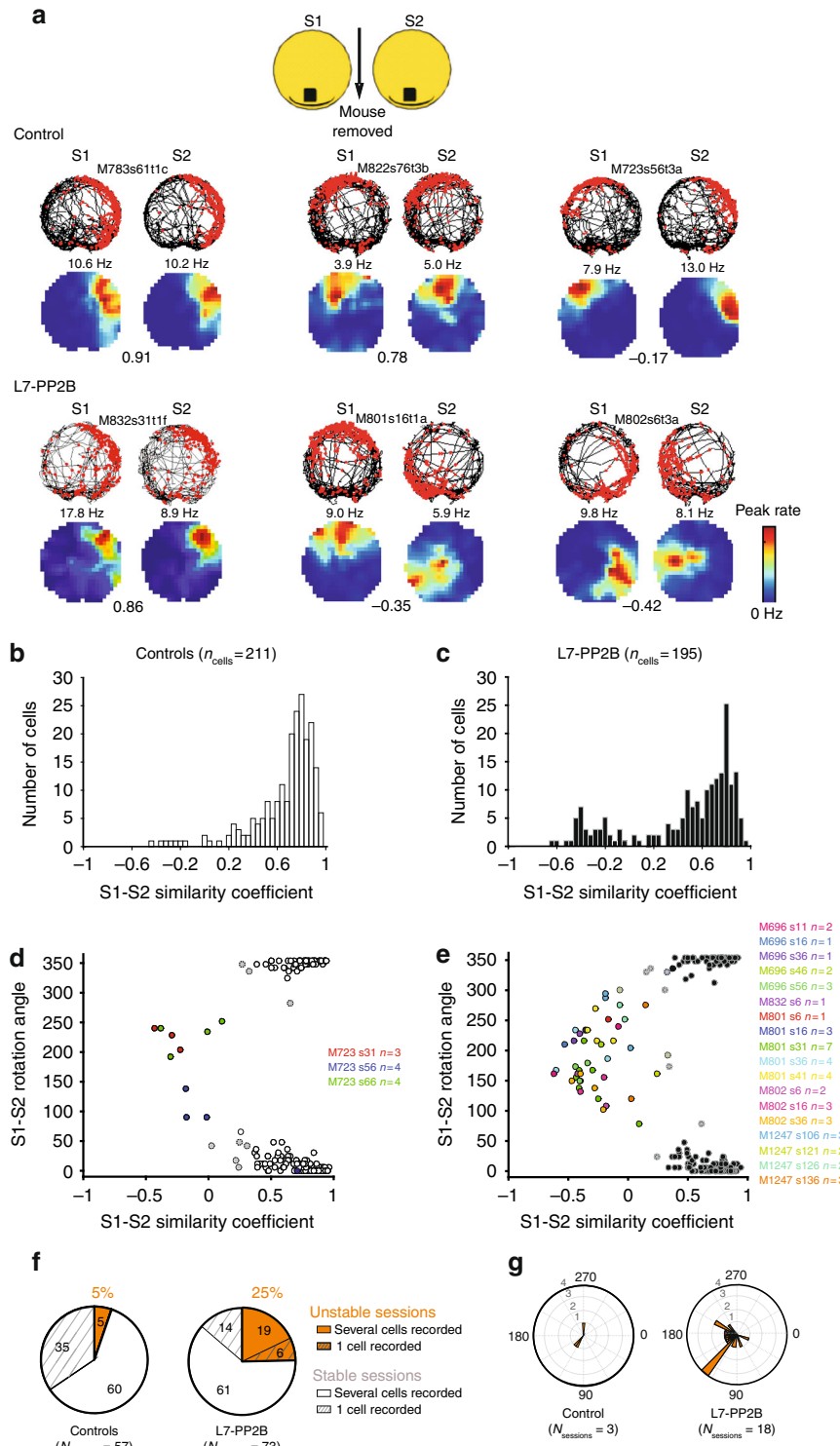

which suggests that the observed instabilities correspond to the coherent rotation of the whole spatial representation. Indeed, during such events, the spatial relationships (angles and distances) between place fields of simultaneously recorded place cells were preserved (Supplementary Fig. 5e–f). Both the mean variation of angles and distance between place fields were below the value obtained on shuffle data (Supplementary Fig. 5e–f). While this instability was sporadic in the control mice (3 recording sessions out of 57, corresponding to 5%), spontaneous rotation of spatial representation occurred more frequently in

mutant mice and were observed in 25% of the recording sessions (18/73 versus 3/57, $p = 0.003$, $\chi^2$ test) among which 14 sessions (19% of total) included multiple place cells (Fig. 1f and Supplementary Fig. 6). Interestingly, several different rotation angles of mouse spatial maps were found between L7-PP2B mice even though the environment remained identical (Fig. 1g and Supplementary Fig. 7). These data suggest that the ability to anchor the spatial representation of a familiar place in a constant reference frame is impaired in L7-PP2B mice. Noticeably, the fact that intra-session stability is normal in L7-PP2B mice

**Fig. 1** The stability of spatial map orientation is altered in L7-PP2B mice. **a** Examples showing 3 place cells recorded during two consecutive identical sessions S1 and S2 in light conditions for control (top) and L7-PP2B mice (bottom). The place cell from the top right illustrates one of the rare examples of a control place cell that was unstable between S1 and S2 (5%) whereas two examples of L7-PP2B place cells displaying instability are shown below (25% of recordings). The animal's path is shown in black and spike locations in red, with a color-coded map of the firing rate below (peak rate is indicated above the map). Cell identity is indicated between trajectories. S1–S2 similarity coefficients are indicated between rate maps. **b**, **c** Distribution of S1–S2 similarity coefficients in the whole population of place cells recorded in control (**b**) and L7-PP2B (**c**) mice. These distributions are significantly different between control and L7-PP2B mice, Kolmogorov–Smirnov, $p < 0.001$. **d**, **e** Classification of recorded cells into stable and unstable neurons with respect to the S1–S2 angle and S1–S2 similarity coefficient in both control (**d**) and mutant (**e**) mice. The classification has been achieved with a K-means algorithm using the two dimension vectors [S1–S2 angle; S1/S2 similarity coefficient] (see Methods). Each dot represents an isolated place cell. Unstable cells are color-coded with a common color for cells recorded in the same session (indicated on the right). **f** Pie-chart illustrating the proportion of unstable recordings for controls (left) and mutants (right) in orange. Unstable recordings occurred more frequently in mutant mice than in controls (18/73 versus 3/57, $p = 0.003$, $\chi^2$ test). A recording episode was classified as unstable if more than 50% of the recorded place cells were classified unstable (see Methods). In a majority of recordings (non-hatched areas) several cells were simultaneously recorded, which allowed subsequent analyses. **g** Polar histogram displaying the median rotation angle calculated for each unstable S1–S2 recording in control and L7-PP2B mice

(Supplementary Fig. 4e) indicates that the ability of the spatial map to correctly anchor to a landmark is altered specifically when the animal has to reset its orientation according to an external landmark (i.e., when it is removed from and placed back into the arena), but once the orientation is settled, it remains stable as long as the animal stays in the arena.

**Altered cue behavior during map rotation in mutant mice.** Analyses of mice behavior at the onset of these trials revealed that spatial map rotations were specifically associated with an increase in cue exploration (Fig. 2).

Investigation of the cue behavior during stable episodes (Fig. 2b, d, f) revealed no main difference between S1 and S2 in control or L7-PP2B mice (all $p > 0.05$ except for the normalized number of entries in the cue zone in control mice, Wilcoxon Signed-Rank test). Given that spatial map rotation almost never occurred in the control mice (only three times in one single individual), analysis of cue behavior during unstable episodes was then focused only on L7-PP2B mice. During spatial map rotation, mutant mice displayed a decrease of the mean distance from the cue ($p = 0.0024$, Wilcoxon Signed-Rank test) and an increase in both the time spent ($p = 0.037$, Wilcoxon Signed-Rank test) and the number of entries in the cue zone ($p = 0.011$, Wilcoxon Signed-Rank test) in S2 compared to S1 (Fig. 2c, e, g). To further decipher if the increase in cue behavior was specific to the spatial map rotation condition, we normalized cue behavior in S2 relative to S1 and compared stables sessions with unstable ones (Fig. 2h–j). S2/S1 ratio analyses confirmed that the increase in cue exploration in session S2 was specifically associated with spatial map rotation in L7-PP2B mice.

During such events, the spatial relationships between mouse representation and the proximal cue are modified. Thus, the increase in cue exploration observed in session 2 could be due to the wrong interpretation that the object has been displaced. To address this point, we further tested the effect of a real cue rotation in a familiar environment on behavior. Since the environment is symmetrical relative to the cue, the cue rotation was performed in front of the animal in order to be detected. Both control and mutant mice displayed the expected increased exploration of the displaced object (Supplementary Fig. 8), similar to the behavior observed following the rotation of the hippocampal spatial map.

**Correct anchoring of L7-PP2B place cells on self-motion cues.** In order to unveil if this deficit resulted from an incorrect anchoring of the hippocampal representation on the visual cue or on self-motion, we further ran a protocol in which the visual cue

was removed after the two familiar sessions S1 and S2 (Fig. 3a, b) and the light switched off. The absence of external visual cues did not affect place cell properties: place field size of L7-PP2B mice remained higher to controls with no effect of the dark sessions (Fig. 3c) and spatial coherence, spatial information content and intra-session stability were preserved in the dark for both controls and L7-PP2B (Fig. 3d–f). In such condition, L7-PP2B mouse general locomotor activity (traveled distance, number of stops, and speed) and center versus periphery exploratory behavior (distance to the wall, percentage of time, and number of entries in the center) was also unaltered (Supplementary Fig. 9, all $p > 0.05$, Mann–Whitney U-test). In line with these results, L7-PP2B mice displayed preserved general sensori-motor properties (locomotor activity, balance and motor coordination) in the dark, a condition during which vestibular input is highly involved (Supplementary Fig. 9). These data strongly suggest that sensori-motor behavior as well as self-motion based hippocampal representation is unaffected in L7-PP2B mice.

Interestingly, when mice re-entered the familiar environment (S5), instability was again observed in L7-PP2B mice (Fig. 4a), and characterized by low S4–S5 similarity coefficients (Fig. 4b) and large rotation angles in L7-PP2B mice (Fig. 4c). The strong coherence between simultaneously recorded cells suggests that these events also correspond to rotation of the whole spatial representation (Fig. 4c), that either shifts back to fit S1 orientation (75% of the cases) or shifts to a new position in 25% of the cases. These data confirm that L7-PP2B mice are impaired in selecting a constant reference frame when they enter a familiar place.

**Increased hippocampal high gamma power in L7-PP2B mice.** To seek insights into the processes underlying place cells instability, we investigated hippocampal local field potential (LFP) activity in control and L7-PP2B mice during two consecutive sessions of free exploration of the familiar environment. LFP activity in theta and low-gamma bands was not different between controls and L7-PP2B mice or between S1 and S2 (Fig. 5b, c, repeated measure ANOVA, all $p > 0.05$ for theta and low gamma band). However, an increase in the power of the high gamma band was observed in session 2 in L7-PP2B mice compared to session 1 and to control mice (Fig. 5b, c, repeated measure ANOVA, genotype, $F_{(1,8)} = 5.3$, $p = 0.05$, session, $F_{(1,8)} = 6.6$, $p = 0.03$; session*genotype interaction $F_{(1,8)} = 6.1$, $p = 0.04$, $p = 0.007$ for both S1–S2 and control-L7-PP2B comparisons, LSD post-hoc test). This result points towards an increase in the computation linked with high gamma in the CA1 region of L7-PP2B mice specifically when mice re-enters the familiar environment.

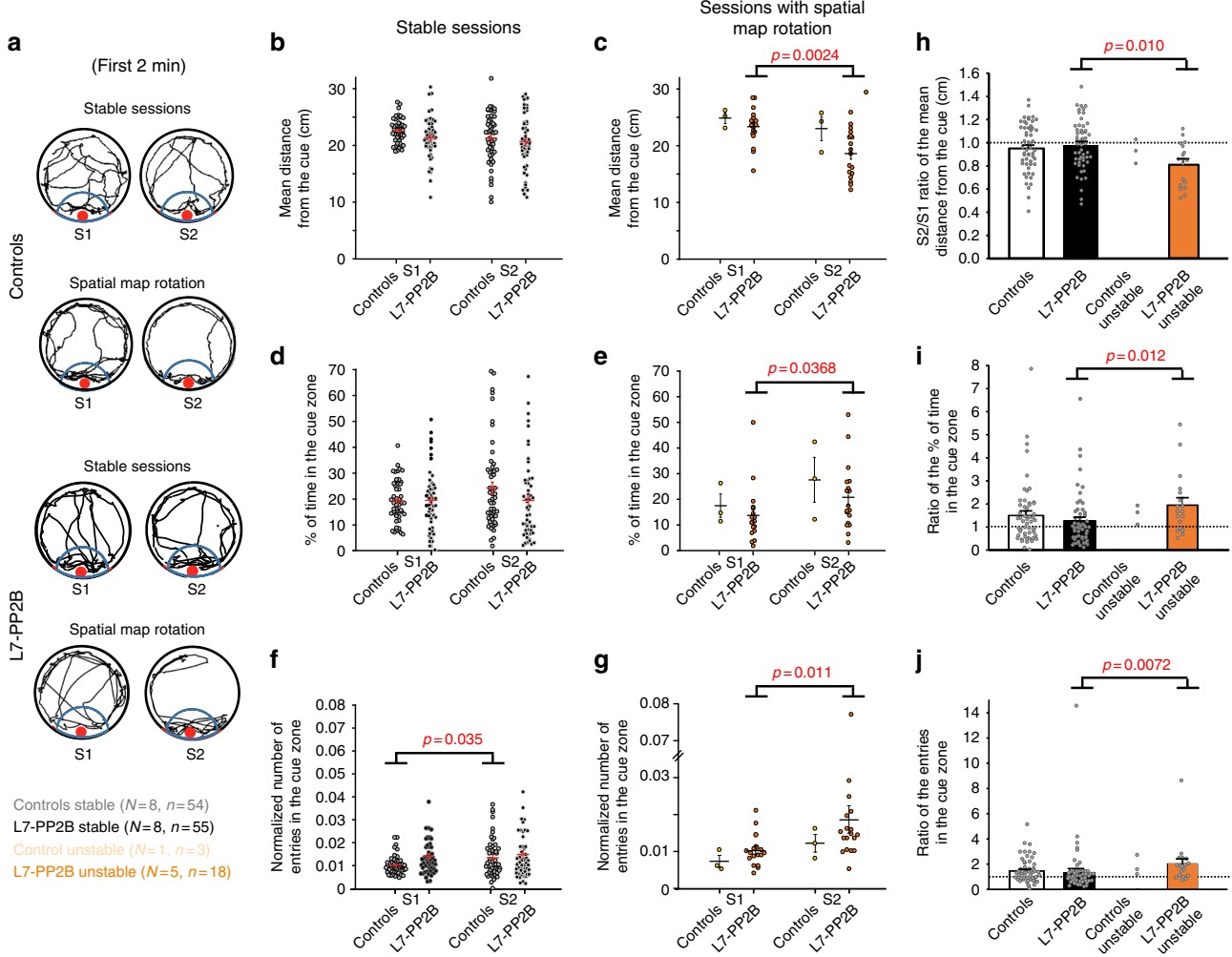

**Fig. 2** Spatial map rotation is associated with increased cue exploration. **a** Examples of control and L7-PP2B mice running tracks during the first 2 min of exploration in consecutives stable sessions, or in sessions associated with spatial map rotation (SSMR). **b**–**g** Scatterplot showing the mean distance from the cue (**b**, **c**), the % of time in the cue zone (**d**, **e**) and the number of entries in the cue zone, normalized by the traveled distance (**f**, **g**) in stable sessions and in SSMR. Basal characterization of cue behavior during the 1st session of stable sessions showed a tendency in L7-PP2B mice to explore more the object than controls (mean distance from the cue, $U = 1681$, $p = 0.011$; % of time spent in the cue zone, $U = 2080$, $p = 0.44$; normalized number of entries in the cue zone, $U = 1349$, $p < 0.001$). An increase in cue behavior was observed in L7-PP2B mice, in S2 relative to S1 during SSMR, but not during stable sessions (mean distance from the cue: stable sessions $Z = 0.82$, $p = 0.82$; SSMR, $Z = 3.24$, $p = 0.0024$; % of time in the cue zone: stable sessions $Z = 0.20$, $p = 0.17$; SSMR, $Z = 2.37$, $p = 0.0368$; normalized number of entries in the cue zone: stable sessions $Z = 0.017$, $p = 1$; SSMR, $Z = 2.77$, $p = 0.011$, Wilcoxon Signed-Rank test). **h**–**j** S2/S1 ratio of the mean distance from the cue (**h**), the percentage of time in the cue zone (**i**), and the normalized number of entries in the cue zone (**j**) in controls and L7-PP2B mice. In L7-PP2B mice, all parameters were modified in SSMR compared to stable sessions (mean distance from the cue, $U = 275$, $p = 0.010$; % of time in the cue zone, $U = 281$, $p = 0.012$; normalized number of entries in the cue zone, $U = 267$, $p = 0.0072$, Mann–Whitney $U$ test). In control mice, no statistical comparison was applied given the low sample size of unstable cells. $p$ values were adjusted for multiple comparisons with a Bonferroni correction. $N$ and $n$ indicate mice and cell number, respectively. Error bars represent S.E.M.

**Impaired allocentric navigation abilities in mutant mice.** Finally, to investigate the functional consequences of impaired cerebellar PP2B-dependent plasticities on spatial navigation, we assessed L7-PP2B mice navigation performances in a Water Maze. We first assessed allocentric navigation in the spatial version of the Morris Water Maze. Since L7-PP2B mice displayed lower swimming speed than control mice (Supplementary Fig. 10b), probably related to their mild motor deficit[33,34], subsequent analyses of the spatial Water Maze were performed using parameters that are independent from speed. The preserved performances found in the cue condition confirmed that visuo-motor ability of L7-PP2B mice is intact (repeated measure ANOVA, $p > 0.05$, Supplementary Fig. 10a). Although L7-PP2B mice did not present a global deficit in their capacity to learn the platform location (Fig. 6a), the fine analysis of their trajectories

revealed alterations of their research behavior. First, the bimodal distribution of swim paths allowed separation of short (direct) and long (indirect) paths using a Gaussian fit (see Methods). This evidenced that the proportion of indirect paths of L7-PP2B mice was significantly higher than controls throughout the 5 days of spatial learning (L7-PP2B 46% (91/199) versus controls 34% (82/238) Fisher exact test, $p = 0.018$, Fig. 6b). Variability in mice performances was then inspected by analyzing the number of transitions between successful (direct) and failed (indirect) trials. We found that L7-PP2B mice performances were indeed more variable as illustrated by the increase in successful/failed trials transitions during the last 2 days of training (i.e., when the performances of control mice reached a plateau) (Fig. 6c).

Additionally, we assessed the initial searching behavior of mice by analyzing their location relative to the hidden platform shortly

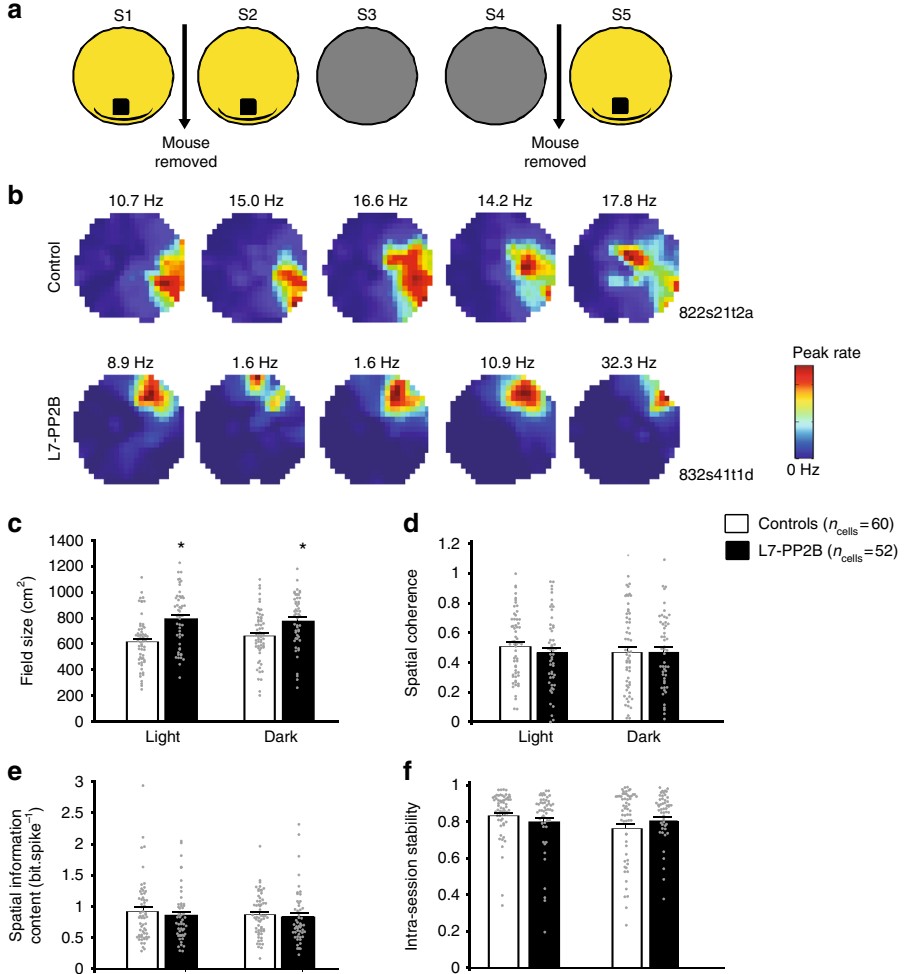

**Fig. 3** Hippocampal place cell properties of L7-PP2B mice are preserved in the dark. **a** Schematic diagram of the protocol used to assess the effect of proximal cue removal on place cell properties. After two consecutive standard sessions (S1–S2), the cue was removed and light was turned off (S3–S4). The mouse was removed from the arena after S4, and S5 was run similarly to S1–S2. **b** Examples of color-coded rate maps showing firing activity of control and L7-PP2B single CA1 pyramidal cells over the five consecutive sessions. **c**–**f** Barplot showing place cell characteristics during familiar sessions (light, S1–S2) or during proximal cue suppression (dark, S3–S4) in control and L7-PP2B mice. Field size was higher in L7-PP2B compared to controls in both light and dark condition (**c**, genotype, $F_{(1,110)} = 16.5$, $p < 0.001$; session, $F_{(1,110)} = 0.63$, $p = 0.43$, session*interaction, $F_{(1,110)} = 3.70$, $p = 0.057$, repeated measure ANOVA) but no difference between light and dark conditions was observed in L7-PP2B mice ($p = 0.46$, LSD post-hoc test). Spatial coherence (**d**, genotype $F_{(1,110)} = 0.44$, $p = 0.51$, repeated measure ANOVA) spatial information content (**e**, genotype $F_{(1,110)} = 0.38$, $p = 0.54$, repeated measure ANOVA), or intra-session stability (**f**, light: $U = 1372$, $p = 0.55$, dark: $U = 1546$, $p = 1$, Mann–Whitney $U$-test with $p$ values adjusted for multiple comparisons) were all similar between control and mutant mice in both sensory conditions. Error bars represent S.E.M.

after they started to swim. Distance to the platform after the initial segment of trajectory was significantly higher for L7-PP2B mice compared to controls (Fig. 6d, for 300 cm, $p = 0.029$, inset: for 200–600 cm, $p < 0.05$, repeated measure ANOVA). Moreover, their search area was less relevant. Indeed, single trial exploration map allowed to identify the zone of focused research (peak of the map, see inset of Fig. 6e). The distance between the platform location and the peak of exploration maps was higher for L7-PP2B compared to controls (Fig. 6e, $p = 0.001$, repeated measure ANOVA). Besides, none of the parameters affected in L7-PP2B mice correlated with speed (Supplementary Fig. 10c–d) indicating that the deficits observed in L7-PP2B mice in the spatial Water Maze do not result from impaired motor performances. In order to unveil if L7-PP2B mice impaired performances result in part from an altered vestibular signal, we tested, in the same behavioral paradigm, control C57/Bl6 mice that have been disoriented before each training session in the Water Maze. In contrast to L7-PP2B mice, passively-disoriented mice abilities to

find the platform were not impaired compared to non-disoriented mice (Supplementary Fig. 11), suggesting that the navigation deficit observed in the L7-PP2B mice do not result from impaired vestibular information per se.

**Impaired self-motion based navigation in L7-PP2B mice.** Finally, we examined the ability of L7-PP2B mice to navigate toward a goal in the dark, i.e., using mainly self-motion information, a function that is altered in mice lacking cerebellar LTD[28]. Mice were first trained to find an escape platform at constant location with a constant departure point in the Water Maze in the light and once the trajectory was learnt (i.e., short and stereotyped) mice performances were further tested in the dark (Supplementary Fig. 12). In the light, L7-PP2B mice rapidly and accurately learned to find the platform as illustrated by the traveled distance, heading, initial orientation and percentage of direct finding that was similar to control (repeated measure

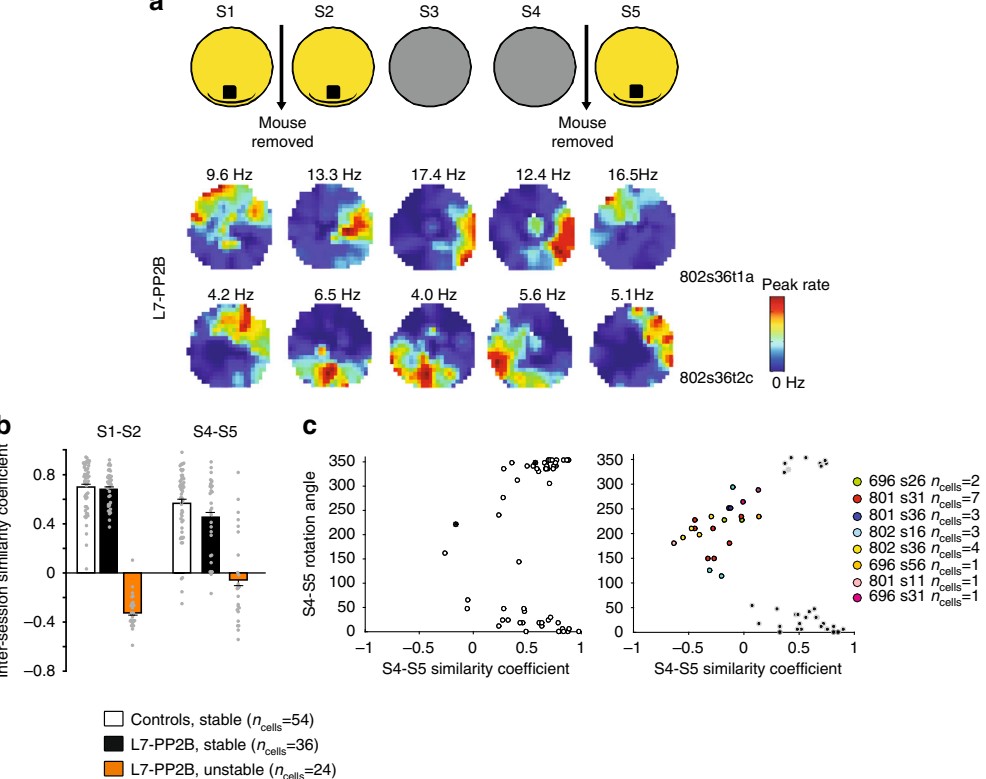

**Fig. 4** Instability occurs upon entry in a familiar environment. **a** Examples of color-coded firing maps of two simultaneously recorded L7-PP2B place cells over the five consecutive sessions, for which instability occurred both at S1–S2 and S4–S5 transitions. **b** Barplot showing inter-session similarity coefficient for control and L7-PP2B place cells at S1–S2 and S4–S5 transitions, i.e., at transitions corresponding to entries into the arena. Cells from stable recordings (white for controls, black for mutants) are separated from cells from unstable recordings identified at S1–S2 transition (orange for mutants). In this protocol, no recording showed S1–S2 instability in control mice. **c** Instabilities at S4–S5 transition occurred only in L7-PP2B mice. Scatter plots showing the distribution of place cells from control and L7-PP2B mice according to their S4–S5 similarity coefficient and S4–S5 rotation angle. Cells from unstable sessions are color-coded, emphasizing the angular coherence of cell ensembles. The few unstable cells observed in the controls come from sessions in which all other simultaneously recorded cells were stable. Error bars represent S.E.M.

ANOVA, all $p > 0.05$, Supplementary Fig. 12). Analyses in the dark indicated that L7-PP2B mice performances were impaired compared to controls as illustrated by the increased in total distance, higher heading, altered initial orientation, and decreased percentage of direct finding (repeated measure ANOVA, total distance, $F_{1,18} = 5.06$, $p = 0.029$; heading, $F_{1,18} = 8.21$, $p = 0.010$; initial orientation, $F_{1,18} = 5.61$, $p = 0.029$, % of direct finding, $F_{1,18} = 6.7$, $p = 0.019$). This deficit was not associated with a change in performance variability (Supplementary Fig. 12c). Together, these data suggest that L7-PP2B abilities to reproduce a short stereotyped trajectory toward a goal in the dark is altered.

## Discussion
The central finding of our study is that L7-PP2B mice exhibit an unstable hippocampal representation that occurs specifically when mice have to orient their spatial map according to the allocentric world.

The susceptibility to spatial map disorientation has been described in rats submitted to a disorientation procedure before each recording session[8]. Remarkably, external disorientation resulted in unpredictable rotation of the place fields relative to the cue card. Thus, the strength of the cue control over place cells was correlated with the rat's learned perception of the stability of the cue. Instability of hippocampal place map has also been previously described in aged rats and it has been suggested that this deficit could result from an unsatisfactory quality of sensory information reaching the hippocampus[13]. Similar interpretation

might apply to L7-PP2B mice deficit as spatial map rotation can also result from a bias in sensory integration.

Previous findings from Schonewille et al.[31] suggest abnormal vestibulo-ocular integration in L7-PP2B mice. This might lead to an unreliable integration of sensory information and subsequently to instability of the spatial representation in the allocentric reference frame. Interestingly, the ability of L7-PP2B mice to maintain a stable spatial representation in the absence of object in the dark (Fig. 3) is coherent with the fact that they maintain the structure of their map (i.e., the spatial relationships between place fields) based on self-motion information even when its orientation is shifted relative to the object.

To explore potential processes underlying place cell instability, we first investigated vestibular function of L7-PP2B mice by assessing sensori-motor properties, exploratory behavior and place cell properties in dark conditions (Supplementary Fig. 9 and Fig. 3). The absence of deficit observed in L7-PP2B in such condition does not suggest that the unstable hippocampal representation of L7-PP2B mice could emerge from a basic alteration of the vestibular function. In addition, the nature of the deficits observed in place cells of L7-PP2B does not fit with the phenotype observed after inactivation of the vestibular system which led to the disruption of location-specific firing in hippocampal place cells[35,36].

To gain insight into mechanisms that could underlie unstable hippocampal representation of L7-PP2B mice, we then investigated hippocampal CA1 LFP activity during free exploration

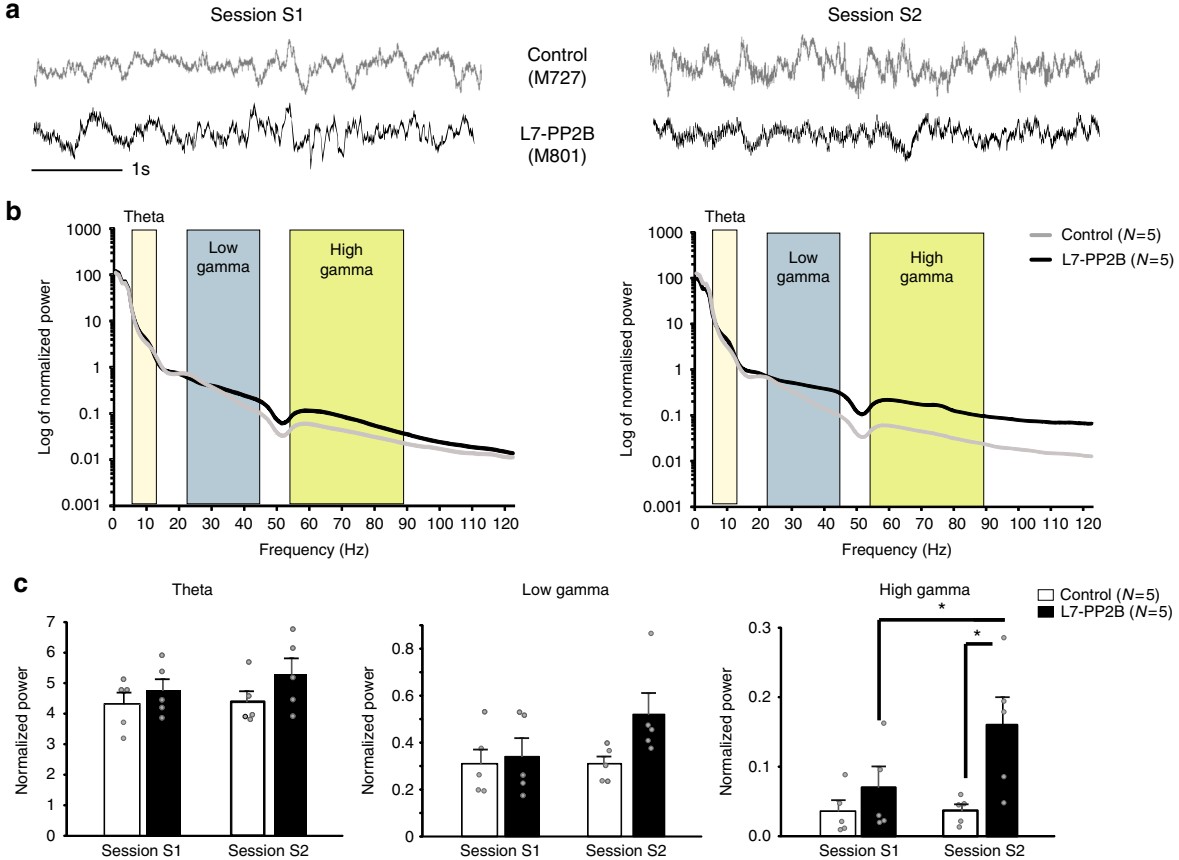

**Fig. 5** L7-PP2B mice have higher high gamma power than controls upon reentry in the arena. **a** Raw traces from one L7-PP2B mice and one control mice during S1 and S2. **b** Average power spectrum normalized by total power in the 1–120 Hz frequency range computed over exploration epochs. **c** Barplots showing the normalized power of the theta (6–12 Hz, repeated measure ANOVA, genotype, $F_{(1,8)} = 1.24$, $p = 0.30$; session, $F_{(1,8)} = 2.44$, $p = 0.16$; genotype*session interaction, $F_{(1,8)} = 0.31$, $p = 0.59$), low gamma (20–45 Hz, repeated measure ANOVA, genotype, $F_{(1,8)} = 1.96$, $p = 0.20$; session, $F_{(1,8)} = 3.32$, $p = 0.10$; session*genotype interaction $F_{(1,8)} = 3.70$, $p = 0.09$) and high gamma (55–95 Hz, repeated measure ANOVA, genotype, $F_{(1,8)} = 5.3$, $p = 0.05$; session, $F_{(1,8)} = 6.6$, $p = 0.03$; session*genotype interaction $F_{(1,8)} = 6.1$, $p = 0.04$; L7-PP2B, S1–S2, $p = 0.007$, controls-L7-PP2B in S2, $p = 0.007$, LSD post-hoc test) bands. *$p < 0.05$ with a LSD post-hoc test, error bars represent S.E.M. $N = 5$ independent mice for each control and L7-PP2B group

of the environment. L7-PP2B mice displayed an increase in high gamma frequency power (55–95 Hz) specifically when they re-entered the familiar environment (Fig. 5). In the hippocampal formation, gamma rhythms may modulate the interaction between substructures. Indeed, low- and high-frequency gamma oscillations are thought to mediate coherence between CA1 and, respectively, CA3 and entorhinal cortex (EC)[37]. Thus, the low gamma/high gamma balance in CA1 may reflect the input dominance toward either a memory-based/CA3 or a contextual/EC computational mode[9]. Here, the increased high gamma computation in L7-PP2B points toward an unbalance in favor of EC inputs towards CA1 that may induce a bias in the integration of external sensory inputs, eventually leading to unstable anchoring of the spatial map to the allocentric frame (Fig. 1).

We previously showed a role of PF-PC LTD in maintaining the location-specific firing of place cells when relying on self-motion cues. Here, L7-PP2B mice spatial code impairment is strikingly different, but might also arise from a default in processing and filtering multimodal sensory information before it reaches the hippocampus. Indeed, a growing body of evidence indicates that the cerebellum integrates multimodal sensory information at the level of granule cell[18,19,38] and Purkinje cell inputs[39], leading to an internal model of sensory consequences of movements that

can be rapidly updated[40]. In particular, cerebellar Purkinje cells activity reflects the ability to transform vestibular signals from an egocentric to an allocentric reference frame[20]. More recently, the existence of translational optic flow-tuned Purkinje cells has also been reported in the same cerebellar area, thus emphasizing the importance of multi-sensory information processing in facilitating the perceptual dissociation of self-motion and object motion[39]. Our results suggest that PP2B-dependent processes may participate in the generation of the cerebellum internal signal, which controls anchoring of the spatial code on its environment. These findings fit with the more recent view that different forms of plasticity dominate different modules of the cerebellar cortex[41–43]. Indeed, control of allocentric processing such as that exerted during spatial integration of visuo-vestibular inputs might be dominated by potentiation of PCs that fire at relatively low baseline simple spike firing rate, whereas egocentric processing such as that occurring during eyeblink conditioning might be dominated by suppression of PCs that fire normally at relatively high firing rate[41,44]. Thus, the present work reinforces and complements our previous findings that the cerebellum participates in the hippocampal spatial code[28]. Here, using the L7-PP2B mouse model in which both synaptic and intrinsic potentiation of Purkinje cells are altered, we reveal an important role of Purkinje cells' computation in maintaining the spatial

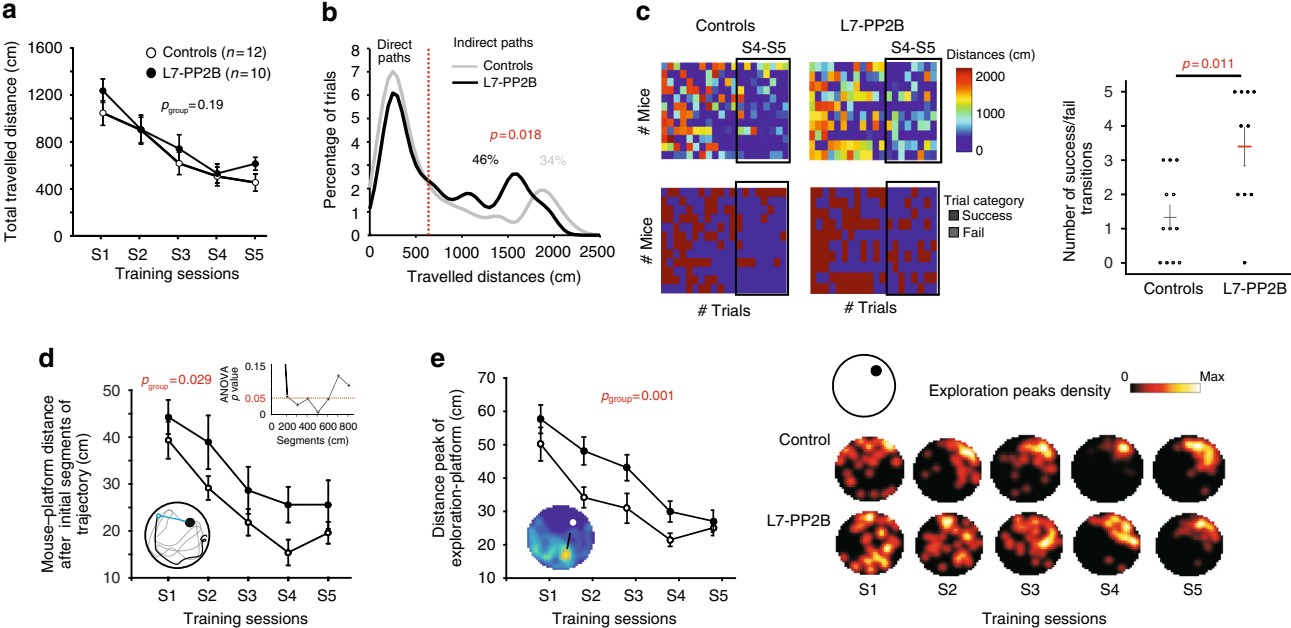

**Fig. 6** L7-PPB mice orientation abilities are impaired in the Morris Water Maze. **a** Performance of control ($n = 12$) and L7-PP2B ($n = 10$) mice improved significantly along the training sessions (repeated measure ANOVA, session $F_{(4,20)} = 16.6$, $p < 0.001$; genotype, $F_{(1,20)} = 1.8$, $p = 0.19$; session*genotype interaction, $F_{(4,20)} = 0.4$, $p = 0.83$). **b** Distributions of traveled distances of all trials for control (gray) and L7-PP2B (black) mice. The bimodal distribution allowed the separation of direct and indirect trials using a Gaussian fit, the threshold is indicated by the red dashed line ($p = 0.018$ with a Fisher exact test). **c** Left, Top. Color-coded representation of traveled distances for all trials. Each line represents the performances of one mouse over the training. Bottom. Same representation with the categorization in successful (direct, blue) and failed (indirect, brown) trials. Right, The number of transitions between successful and failed trials is higher in L7-PP2B mice compared to controls ($U = 22$, $p = 0.011$, Mann–Whitney $U$-test) at the end of training (S4 and S5, indicated by the black frame on the left panel). **d** Mouse initial orientation is evaluated by the distance between mouse position and platform location after a 300 cm initial trajectory (repeated measure ANOVA genotype, $F_{(1,20)} = 5.6$, $p = 0.029$; session, $F_{(4,20)} = 13.5$, $p < 0.001$, session*genotype interaction, $F_{(4,20)} = 0.2$, $p = 0.92$). The cartoon illustrates a trajectory in gray, the initial segment in black and the mouse–platform distance in blue. Inset: ANOVA $p$ values for the same analysis performed for different lengths of initial trajectory segments showing that $p < 0.05$ from 200 to 600 cm lengths. **e** Left. Mean distance between the peak of exploration (corresponding to mouse search area) and the platform location (repeated measure ANOVA, genotype $F_{(1,20)} = 14.8$, $p = 0.001$; session $F_{(4,20)} = 20.1$ $p < 0.001$, genotype*session interaction, $F_{(4,20)} = 20.1$, $p = 0.54$). The cartoon illustrates a map showing the platform (white circle), the peak of exploration (yellow), and the peak–platform distance (arrow). Right. Heat maps showing the spatial distribution of exploration peaks along the training. The cartoon indicates the platform location (black disk) in the pool (top left). See Supplementary Fig. 13 for a detailed description of the analyses. Error bars represent S.E.M.

representation of a familiar environment in a stable allocentric reference frame.

Interestingly, L7-PP2B mice navigation performances were impaired in both allocentric and self-motion conditions, suggesting that a deficit in cerebellar plasticity might impact spatial behavior independently of the sensory context (visual or self-motion). These results are in light with our previous findings on L7-PKCI mice, which also displayed altered path optimization toward a goal in both allocentric[29] and self-motion conditions[28]. In contrast, cerebellar influence on hippocampal activity seems to operate through different forms of plasticities depending on the sensory context. Hippocampal spatial code of L7-PKCI mice is preserved in allocentric condition but altered when using self-motion[28]. Interestingly, L7-PP2B mice present the inverted dichotomy with an altered spatial hippocampal code during allocentric conditions but preserved self-motion based representation.

In conclusion, the cerebellar cortex may participate in the building of an output signal combining external and self-motion information crucial for stabilizing the hippocampal code and optimizing paths during goal-directed navigation. The major differences in hippocampal place cells activity subsequent to the absence of PKC-dependent and PP2B-dependent mechanisms emphasize the specificity of the contribution of each plasticity to information processing in the cerebellar cortex. It demonstrates

the critical role of PP2B-dependent cerebellar plasticities in sensory information processing used to stabilize the hippocampal spatial map orientation in an allocentric reference frame.

## Methods

**Subjects**. All animals were bred in a C57BL/6 mouse strain background and were housed in standard conditions (12 h light/dark cycle light on at 7 am, water and food ad libitum). All data were obtained during the light phase, in compliance with the European Commission directives 219/1990 and 220/1990 and approved by the Comité d'Ethique En Expérimentation Animale Charles Darwin C2EA-05 (Jerôme Yelnik, project 00896.01). Experiments were carried out in blind conditions with respect to the genotype.

For electrophysiology, eight transgenic L7-PP2B mice and eight wild-type littermate controls were recorded in this study. In L7-PP2B mice, the selective deletion of PP2B in Purkinje cells was obtained using the Cre-loxP-system. L7-PP2B mice display normal cerebellar histology and an intact induction of LTD at the parallel fiber-Purkinje cell (PF-PC) synapse. However, no LTP could be induced at this synapse after stimulation of parallel fibers in contrast to the reliable potentiation induced in their controls[31]. Noticeably, L7-PP2B mice also display defects in the baseline excitability and intrinsic plasticity of Purkinje cells. At the behavioral level, L7-PP2B mice are impaired in motor learning both in the adaptation of the vestibulo-ocular reflex and in the eyeblink conditioning task[31]. Control mice included 5 (flox/flox, +/+ ) mice, 1 (+/+ , cre/+) mice, and 2 (+/+ , +/+) mice.

For goal-directed navigation tasks, 33 transgenic L7-PP2B (flox/flox, cre/+) mice and 47 wild-type littermate controls participated in this study. Control mice included 36 (flox/flox, +/+) mice, 6 (+/+ , cre/+) mice, and 5 (+/+ , +/+) mice. Mice were used between 3 and 6 months of age.

**Surgery**. Mice were implanted at 2.5 months of age and recorded until 6 months. The mice were housed individually starting from 1 week before surgery. Mice were deeply anesthetized by injection of Xylazine (10 mg kg⁻¹) and Ketamine (100 mg kg⁻¹) and then placed on a stereotaxic apparatus. Levels of anesthesia were monitored regularly by testing toe and tail pinch reflexes. An implant included 4 tetrodes (each consisting of 4 twisted 25-μm Formvar-insulated nichrome wires) inserted into a single 25-gauge guide cannula. Each wire was attached to a pin of an EIB-18 (Electrode Interface Board-18, Neuralynx Inc., USA). Wire impedance was lowered to 200 kΩ using a gold plating solution enriched with polyethylene glycol[45]. Three steel electric wires were used as reference electrodes and connected as well to the connector. Reference wires were positioned at the brain surface above the right or left cerebellar cortex (1 and 2 wires, respectively). A midline incision of the scalp was made, and the skin and muscle were carefully retracted to expose the skull. Tetrodes tips were positioned above the right hippocampus (AP, −2.0; ML, −2.0, relative to the bregma and DV, −0.9 mm relative to the brain surface). The microdrive was secured to the skull using dental cement (SuperBound C&B, UNIFAST Trad) and protected by successive layers of plastic paraffin film (Parafilm) and dental cement. After implantation, the electrodes and cannula were progressively lowered in the brain by screwing the screw into the Teflon cuff.

**Recording protocol**. Starting 5 days after surgery, the activity from each tetrode was screened daily while the mice explored the recording cylinder. If no waveform of sufficient amplitudes was detected, the tetrodes were lowered by 60 μm steps until hippocampal units could be identified. Signals were amplified 2000 times, filtered (band-pass 0.6–9 kHz) using amplifiers and processed with the animals' position signals using Datawave SciWorks acquisition software. Waveforms of identified units were sampled at 32 kHz and stored. Along with unit data, hippocampal pyramidal LFPs (filtered between 1.0 and 475 Hz) were recorded from one electrode using a 11.67 kHz sampling rate. Video recording was performed using a CCD camera (Mintron, PAL) fixed to the ceiling above the arena and the mouse position was tracked by contrast, at a sampling rate of 25 Hz.

Hippocampal activity was recorded while animals explore a white circular arena (50 cm diameter, 30 cm high, made of white polypropene and the floor covered with white linoleum). The arena was placed on a white elevated platform in the center of the recording room (1.90 × 1.40 m) and surrounded by a black circular curtain. The intramaze cue included a prominent light blue plastic card (29 × 21 cm) placed on the wall of the arena and a white plastic bottle (16 cm high) located against the card center. The apparatus was lit with two lights symmetrically attached to the ceiling, providing a homogeneous light over the whole arena (50 lux), as well as an infrared light to allow tracking in the dark. A white noise generator was also installed to the ceiling in order to mask potential auditory cues (75 dB). Unlike rats, no food motivation was used to stimulate exploratory activity throughout the recording sessions.

After identification of recordable cells, two consecutive 12-min standard recording sessions were run (i.e., with light and the intramaze cue). Before each session, the mouse was carried wrapped in a towel, preventing it from seeing the environment and the experimenter performed one rotation to mildly disorient the animal. For each recording session, mice were systematically placed at the same position in the arena, in front of the object. Between the two sessions the mouse was returned to its home cage (4 min inter-session intervals). Each time that the mouse was removed between sessions, the arena and the cue were cleaned with soapy water to homogenize potential odor-cues. Several sessions were experienced by each mouse (ranging from 2 to 16) and the location of the cue was always at the same position between different S1–S2 sessions. Tetrodes were lowered by 15–30 μm steps between 2 days of recordings.

For some cells (randomly chosen), a 3-sessions cue removal protocol was run immediately after, in order to examine the influence of external information on place cell firing. The mouse stayed in the arena, the light was turned off, the cue (card + object) was removed and sessions 3 and 4 were run. After the mouse returned to its home cage, a last standard session (S5, similar to S1 and S2) was run to check that, whatever the changes in cell firing observed during the cue manipulation sessions, the initial firing pattern could be restored. In the dark condition, the mouse position was tracked by contrast, using infrared light.

**Analysis of electrophysiological recordings**. Spike sorting and cell classification: simultaneously recorded units were clustered manually using Plexon Offline Sorter software. Clustering on each tetrode was based on all possible X–Y combinations of characteristic features including maximum and minimum spike voltage, time of occurrence of maximum spike voltage, principal components (1–3 usually). Auto-correlograms were used to select only well-discriminated clusters (absence of detected event in the refractory period). Cross-correlogram and auto-correlogram comparisons allowed checking unit identity. Unit activity separation for the successive sessions in the recording sequence was based on the initial cluster cutting created for session 1.

Pyramidal cells and interneurons were distinguished on the basis of their average firing rate, spike shape (spike length, initial slope of valley decay), and firing pattern (auto-correlogram). Among the pyramidal cells that were recorded, only cells with clear location-specific activity, i.e., with a place field, were categorized as place cells and included in the data set.

All data were speed-filtered, i.e., only epochs with instantaneous running speeds of 2.5 cm s⁻¹ or more were included. Final positions were defined by deleting position samples that were displaced more than 100 cm s⁻¹ from the previous sample (tracking artefacts), interpolating any missing position points with total durations less than 1 s, and then smoothing the path with a Robust Lowess method (linear fit, span = 11)[46]. To characterize firing fields, the position data were sorted into 2.5 cm × 2.5 cm bins. Data from a particular session were only accepted for analysis if more than 65% of the bins were covered by the animal.

Rate maps and analysis of place cells: firing rate distributions were determined by counting the number of spikes in each 2.5 cm × 2.5 cm bin as well as the time spent per bin. Maps for number of spikes and time were smoothed individually using a boxcar average over the surrounding 5 × 5 bins. Weights were distributed as follows:

box = [0.0025 0.0125 0.0200 0.0125 0.0025;
0.0125 0.0625 0.1000 0.0625 0.0125;
0.0200 0.1000 0.1600 0.1000 0.0200;
0.0125 0.0625 0.1000 0.0625 0.0125;
0.0025 0.0125 0.0200 0.0125 0.0025]

Firing rates were determined by dividing spike number and time for each bin of the two smoothed maps.

A place field was defined as a set of at least 10 contiguous pixels with a firing rate above the overall mean firing rate. Color-coded firing rate maps were then constructed for each session to visualize the positional firing distribution. In such maps the highest firing rate is coded as red, the lowest as blue, intermediate rates are coded as green, yellow, and orange, and unvisited pixels are shown in white.

In addition to the qualitative description of the place cell firing provided by the maps, several numerical measures were used to analyze spatial firing of place cells including the (i) mean field firing rate; (ii) field peak firing rate; (iii) coherence, which measures the local smoothness of firing rate contours; (iv) spatial information content, which expresses the amount of information conveyed about spatial location by a single spike[47]; (v) intra-session stability; and (vi) inter-session stability.

Coherence was estimated as the first order spatial autocorrelation of the place field map, i.e., the mean correlation between the firing rate of each bin and the averaged firing rate in the 8 adjacent bins[48]. Spatial coherence was calculated from unsmoothed rate maps.

For each cell, the spatial information content in bits per spike was calculated as information content: $I = \Sigma_i (\lambda_i/\lambda) \times \log_2 (\lambda_i/\lambda) \times P_i$ where $\lambda_i$ is the mean firing rate in each pixel, $\lambda$ is the overall mean firing rate, and $P_i$ is the probability of the animal to be in pixel $i$ (i.e., occupancy in the $i$-th bin/total recording time).

The inter-session stability, i.e., the spatial correlation between consecutive sessions was estimated for each cell by correlating the rates of firing in corresponding bins of the pair of smoothed rate maps.

Intra-session stability was estimated by computing spatial correlations between rate maps for the first and second halves of the trial.

The place field angular shift between 2 sessions was measured by performing a cross-correlation as the firing rate array of the first session was rotated in 6° steps relative to the firing rate array of the second session. The angle associated with the highest correlation was taken as the rotation angle of the place field between the 2 sessions.

A cell was classified as stable or unstable based on its similarity coefficient and its rotation angle between standard sessions S1 and S2. The K-means algorithm was used to define two subpopulations using the two dimension vector [S1–S2 angles; S1/S2 similarity coefficients]. Because the two clusters were contiguous, the classification of cells close to the threshold defined by the two clusters was uncertain. Distances from the centroids' barycenters were computed and the 5% cells the farthest away from the centroid were classified as uncertain and not included in subsequent analyses. This led us to exclude 10 control cells and 11 mutant cells from this analysis. Since the K-mean algorithm requires a priori definition of the number of clusters, we used the elbow method[49] to check a posteriori that a 2 cluster separation was the most appropriated.

A session was classified as unstable when more than 50% of its place cells were unstable. A session was classified as stable when all the place cells were stable. The few sessions (2 in controls, 1 in mutants) with a percentage between 0% and 50% corresponding to sessions with several simultaneously recorded stable cells including one cell classified as unstable but visually stable were not included in the analyses. The few unstable recordings containing one place cell were included in the proportions for pie charts (Fig. 1f) and taken into account for behavioral analyses (Fig. 2).

Analysis of spatial relationships and angular coherence during instability episodes: for all possible pairs of cells of a recording session, the distance between their barycenters was computed for S1 and S2. The variation in distance $\Delta d$ (absolute value) between S1 and S2 was computed, averaged over all pairs of the recording session, and then over all recording sessions. This value was compared to chance level by a permutation test. To do so, for each set of place field barycenters, a corresponding set of barycenters (matched in sample size) was selected from shuffle data and used to compute a random variation in distance (random $\Delta d$). This was done for all sets of place field barycenters and allowed to compute a mean random $\Delta d$. This permutation procedure was repeated 200 times. Mean $\Delta d$ computed on data was then compared to the distribution of mean random $\Delta d$.

To examine the angles between the place field barycenters, a similar approach was used. For a given pair of barycenters A and B, the angle AÔB was computed, O being the center of the arena. The variation in angle AÔB between S1 and S2 was computed for all possible pairs of cells of a recording session, averaged for the recorded session, and then over the recording sessions. Similarly, this value was compared to chance level by a permutation test. For each set of place field barycenters, a corresponding set of barycenters was randomly selected for S1 and S2, variations in angles AÔB were computed. This was done for all sets of place field barycenters and allowed to compute a mean random variation in angle (mean random ΔΘ). This permutation procedure was repeated 200 times. Mean ΔΘ computed on data was then compared to the distribution of mean random ΔΘ.

First for each of these episodes, the rotation angles of all place cells were determined (i.e., the place field angular shift, see above). The mean rotation angle and the circular standard deviation (SD) were then computed for each set of simultaneously recorded place cells. This circular SD was then averaged over the experiments and compared to chance level by a permutation test. To do so, for each set of rotation angles, a corresponding set of angles (matched in sample size) was selected from shuffle data and used to compute a random SD. This was done for all sets of rotation angles and allowed to compute a mean random SD. This permutation procedure was repeated 200 times. Mean SD computed on data was then compared to the distribution of mean random SD.

For LFP analysis, the raw data was first notch filtered around 50 Hz to remove noise from electrical current. Power spectrum density (PSD) was calculated between 1 and 150 Hz using the multitaper method of the Chronux toolbox[50] in Matlab. PSD was measured using 1 s windows and averaged only when the speed was superior to 2.5 cm s$^{-1}$ and using 5 tapers.

For mice behavior during exploration of the familiar environment, to analyze mice behavior during exploration of the recording arena, the traveled distance, number of stops, mean running speed, distance from wall, time spent in the center and the number of entries in the center normalized by the distance were computed. These general exploration parameters were analyzed over the 12-min session. The exploration of the cue (object + card) was evaluated by the percent of time spent in the object zone (a restricted zone 5 cm around the object), the number of entries in this zone normalized by the traveled distance, and, the mean distance to the object (averaged over the track).

**Histology**. Tetrode positions were checked in seven out of eight L7-PP2B and four out of eight control implanted mice. The mice received an overdose of a Xylazine–Ketamine mix and were perfused intracardially with saline and 4% paraformaldehyde. The brains were extracted and stored in paraformaldehyde, and coronal sections (50 μm) were cut using a vibratome. All sections were mounted on glass slides and stained with cresyl violet. A light microscope fitted with a digital camera was used to determine tetrode placement in the CA1 pyramidal layer.

**Goal-directed navigation tasks**. Before undergoing a navigation task in the Water Maze, all mice were submitted to a general sensori-motor evaluation in either light or dark conditions[28,29]. This included an evaluation of anxiety in the elevated plus maze, which consists in a cross-shaped maze made of black perspex with a central zone (8 × 8 cm) facing closed and open arms (24 × 8 cm, surrounded by 25 cm walls made of gray perspex), elevated to a height of 50 cm. The percentage of time spent and the number of entries in the open arms was measured. An entry was considered valid when the 4 paws were present in the arm. The test lasted for 5 min. Spontaneous locomotor activity was evaluated in a squared arena made of gray perspex (45 × 45 cm) surrounded by red plexiglass walls (30 cm height). Mice were first positioned at the center of the arena and were allowed free exploration for 10 min. Walking time, traveled distance and rearing were then analyzed. Dynamic balance was evaluated using the horizontal rod test. The aim of this test was to estimate the mouse's ability to maintain its balance while in motion. The apparatus consisted of a horizontal rod (50 cm long, 5 cm in diameter) covered with sticking plaster providing a good gripping surface. It was located 80 cm above a soft carpet to cushion the eventual fall of the animals. Both ends of the beam were limited by white altuglass disk (50 cm in diameter). The mouse was placed on the middle of the rod, its body axis perpendicular to the rod long axis. During the test, the time before falling, the distance traveled and the walking time were recorded. The test ended when the animal fell or after 180 s. Static balance was quantified by using an unstable platform. The aim of this test was to evaluate the capabilities of the mice to maintain balance when their displacements were limited. The apparatus consisted of a circular platform (diameter 8.5 cm) made of gray perspex, fixed at its center on a vertical axis (1 m high and 3 cm in diameter) and which could tilt by 30° in every direction. The mouse was placed on the middle of the board (horizontal situation) and the latency before falling (cut-off: 180 s) and the number of slips (when at least one paw was out of the circumference of the platform) were measured. Motor coordination was assessed in an hole-board which consisted of an experimental box made of transparent altuglass (32 × 32 × 25 cm), in which the floor board made of white altuglass has 36 holes (2 cm in diameter, 2 cm deep) arranged in a 6 × 6 grid. The mouse was placed in the middle of the board and its behavior was recorded during 5 min. The walking time and the frequency of stumbles, a measure of motor coordination were calculated. Motor adaptation was assessed using a rotarod task, either on a constant speed (5 and 10 rpm, 4 trials each) or on an accelerating protocol (4 trials)[28,29].

The spatial Morris Water Maze task: mice were trained in a circular water tank (150 cm in diameter, 40 cm high) to find an escape platform (10 cm of diameter) hidden 0.5 cm below the surface of the water at a fixed location. The pool contained water (21 ℃) made opaque by the addition of an inert and nontoxic product (Accusan OP 301). Both the pool and the surrounding distal cues were kept fixed during all experiments. Mice underwent a two-phase training protocol: cue training (5 days) followed by spatial training (5 days). During the cue training, an object (11 cm high) was used to mark the platform (which was randomly placed at different locations across trials), and the pool was surrounded by blue curtains to occlude extramaze cues. During spatial training, prominent extramaze cues placed around the testing room enabled the animals to learn the platform's location (fixed over training). Training consisted in one training session per day, four trials per session. The starting position (North, East, West, or South) was randomly selected with each quadrant sampled once a day. At the beginning of each trial, the mouse was released at the starting point and made facing the inner wall. Then, it was given a maximum of 90 s to locate and climb onto the escape platform. If the mouse was unable to find the platform within the 90 s period, it was guided to the platform by the experimenter. In either case, the mouse was allowed to remain on the platform for 30 s. Data acquisition was performed at a frequency of 25 Hz using the SMART® video recording system and tracking software. Data processing was automated using NAT (Navigation Analysis Tool), a Matlab-based software (NAT: Navigation Analysis Tool) developed in our laboratory[51]. The traveled distance to reach the platform and the average speed were analyzed. To analyze the proportion of successful trials during training, we used a Gaussian fit on the distribution of traveled distances to separate successful (short) and failed (long) trials. Since the Gaussian fit was centered on the successful trial distances the threshold was defined as the distance above which more than 90% of the distribution was no longer explained by the fit (note that a bimodal Gaussian fit could not be used since the distribution of failed trials could not be fitted by a Gaussian curve). Variability in mouse's performances was inspected by analyzing the number of transitions between successful and unsuccessful trials at the end of training (sessions 4–5, when the platform location is known). In order to evaluate the relevance of their initial research behavior in the pool, the mouse–platform distance was computed after the initial segment of the trajectory (300 cm, but see inset in Fig. 5d for path length of 200–600 cm). The relevance of their focused research behavior was evaluated by computing single trial occupancy maps and identifying the peak of their research area. The distance between the peak of exploration and the platform location was compared between groups. To illustrate the spatial distribution of these focused research areas, exploration peak positions were collected to compute a density map (Supplementary Fig. 13).

For the disorientation task, in a separate experiment, 22 C57-Bl6 adult male mice underwent a disorientation procedure while performing the Morris Water Maze Task. The intent of the procedure was to disrupt the mouse's internal sense of direction (i.e., mainly dependent on vestibular information) so that mice orientation mainly relies on visual distal cues. Before each session mice were taken from their home cage to a large (20 cm diameter, 13 cm high) circular covered box and carried up and down in an irregular fashion into the experimental room and around the pool a random number of time (between 1 and 3 turns). In addition, the box was gently randomly rotated (between 1 and 5 turns). The mouse was taken out of the box on top of the starting point and was placed directly into the pool. The control group (N = 20 mice) did not undergo the disorientation procedure. These mice were carried from their home cage directly to the starting point of the pool.

For the self-motion based navigation task, the ability to navigate using self-motion cues was assessed by training the animals in a modified version of the Morris Water Maze task[28]. An escape transparent platform (10 cm diameter), was hidden 1 cm below the water surface, and placed in the pool at a fixed position (30 cm from the edge of the tank). A proximal cue was placed on the top of the platform. During all sessions a trapezoidal-shaped arm (30 cm long, 6 cm high and 9.5 cm width at the pool edge versus 3.5 cm at its distal extremity), oriented towards the platform, was used as departure zone. The departure arm and the platform positions were identical throughout the experiment. Spatial navigation using self-motion cues was evaluated by submitting the animals to 3 sessions per day with 2 h interval between sessions and 30 s inter trial interval. A pretraining of 13 sessions was initially performed on both L7-PP2B and control mice. Twelve training sessions were then conducted in two different lighting conditions (light and dark). The pool was normally lit by 4 ceiling spotlights (25 W) arranged symmetrically relative to the center of the maze. In this condition, the light level in the center of the pool and at the water surface was 84 lux and was homogenous across the room. In the dark condition (0 lux), the ceiling lights were switched off and were offset by two infrared sources. The light was switched on again when the animal reached the platform or after 60 s. After each session, the animals were warmed up under a red light behind the curtains. All the different navigation parameters were analyzed during light and dark conditions.

**Statistics**. All statistics were carried out with Matlab statistics toolboxes and the STATISTICA software. For all statistical tests, data were tested for normality and homogeneity of variances using a Kolmogorov–Smirnov test to ensure they met the necessary assumptions before proceeding to analysis. Statistical tests included Mann–Whitney U-test, Student's t-test, Wilcoxon Signed-Rank tests, Chi-square

tests, Fisher tests, and repeated measure analyses of variance (ANOVA). Distributions of similarity coefficients were compared by a Kolmogorov–Smirnov test. Circular statistics were made using Circstat, a Matlab toolbox for circular statistics[52]. When multiple comparisons were applied, $p$ value was adjusted with a Bonferroni correction. The significant threshold was fixed at 5% ($p < 0.05$ was significant). All data are presented as mean ± S.E.M. unless stated otherwise.

**Reporting summary**. Further information on research design is available in the Nature Research Reporting Summary linked to this article.

## Data availability
The data that support the findings of this study are available from the corresponding author upon reasonable request.

## Code availability
The custom codes that support the findings of this study are available from the corresponding author upon reasonable request.

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

## Acknowledgements

The authors thank Gregory Lafon, Julien Schmitt, and Aurélie Watilliaux for technical support; Nadir Benlismane and Audrey Goulian for animal care; and Aurélie Watilliaux for mouse colony supervision and genotyping. The authors thank Glenn Dallérac and the CeZaMe team members: Mehdi Fallahnezhad, Anne-Lise Paradis, and Thomas Watson for their constructive comments on the manuscript. The authors also thank the Moser's lab for sharing part of their Matlab codes. This work was supported by the Institut DEQ20120323730-France, by the National Agency for Research ANR-REG-071220-01-01-France (L.R.-R.), and by the Institut Universitaire de France (C.R.). This work also received support under the program Investissements d'Avenir launched by the French Government and implemented by the ANR, with the references, PER-SU (L.R.-R.) and ANR-10-LABX-BioPsy (L.R.-R.). The group of L.R.-R. is member of the Labex Bio-Psy and ENP Foundation. Labex are supported by French State funds managed by the ANR within the Investissements d'Avenir programme under reference ANR-11-IDEX-0004-02. L.J.M. was funded by the Ministère de l'Enseignement Supérieur et de la Recherche, France and FRM. C.I.D.Z.. is funded by the Dutch Organization for Medical Sciences, Life Sciences, and Social and Behavioral Sciences, and ERC-adv and ERC-POC.

## Author contributions

L.R.-R. designed the project and coordinated the grants. C.R. and L.R.-R. designed and coordinated the experiments. J.M.L. participated in setup and program development with the help of F.J. C.I.D.Z. provided the conditional knockout mice. J.M.L., C.R. and L.R.-R. designed the electrophysiological and behavioral analyses. J.M.L., J.V. and C.R. collected and analyzed the data. L.T. analyzed the LFP. L.R.-R., C.R. and J.M.L. wrote the manuscript with comments from C.I.D.Z.

## Additional information

**Competing interests:** The authors declare no competing interests.

