## [Peer Review File · Nature Communications]

Reviewers' comments:

Reviewer #1 (Remarks to the Author):

In this manuscript, the authors used L7-PP2B knockout mice to examine how the loss of long-term potentiation and intrinsic excitability impact hippocampal spatial representations. They report decreased stability of place cells in a cued circular environment in L7-PP2B mice compared to control mice. This instability was bi-modal; all place cells in a session either remained stable or remapped between S1 and S2. Unstable sessions were associated with increased exploration of the cue. Visual cue removal did not affect stability but reintroducing the mice to the environment again caused instability. Lastly – the L7-PPB mice exhibit deficits in the self-localization abilities in the spatial version of the Morris Water Maze. Dissecting the role of the cerebellum in generating stable neural representations of space is an important topic and the author's use of a unique knockout animal to target a specific feature of cerebellar processing is novel and interesting. However, I have serious concerns about the degree to which the phenotype they observed is robust, the level of clarity regarding the potential mechanism underlying their observations and how the behavioral data fits with the effects they observed in the hippocampal neural data.

1. Generally speaking, I was not convinced by the electrophysiological effects presented. The knockout of PP2B in Purkinje cells appears to marginally increase the amount of sporadic place cell remapping that occurs across behavioral sessions (3/53 sessions in wildtype compared to 11/61 sessions in knockout mice). The numbers of 'unstable sessions', as well as the number of knockout mice that compose these sessions (n = 3), are both very low.

2. In addition to the low n, I'm not sure how to interpret the bi-modal effect the authors observe on place cell stability. If there is a significant deficit in the ability of these animals to maintain a stable allocentric reference frame, why was the change in stability observed in so few sessions? Instead, this suggests to me, there could be some other underlying effect causing the sporadic remapping. For example, in Figure 2, it appears that coverage of the environment was not as uniform in sessions that were picked up as being unstable. A systematic coverage distance would not be picked up by just calculating traveled distance (as the mice could travel a very long distance but never enter the center of the arena). If the coverage is low in the center of the box, it could be that place cells appear to shift simply because the mouse failed to uniformly sample the entire environment and thus, aberrant spikes were defined incorrectly as place fields. Or perhaps the animals were placed in the environment from very different positions or angles on certain S1 to S2 trials compared to other trials.

3. The take-home message of the paper is also not clear to me. The authors report a bi-modal effect, where knockout mice had slightly more sessions in which place cells coherently remapped. However, how this effect relates to LTP in Purkinje cells, as well as how to relate this effect to a difference in short versus long path lengths in the Morris Water Maze is not clear to me.

4. The mechanism underlying the place cell instability is not clear. One potential way the paper could be improved would be to add simultaneous recordings of head direction cells. The presence of drift or remapping in the head direction cell population might at least point to a potential interpretation of the effects reported in place cells.

5. For Figure 2 – the increased exploration of the cue in sessions where the rotation occurs could indicate that the animals think the object has been displaced (or is novel). If you use the same environment but rotate the cue – do KO mice explore the cue more than WT mice?

6. Can the authors dissociate the impact of the change in place cell field size versus stability on the behavioral deficits they observed in knockout animals?

7. Given that L7-PP2B mice show instability only in some trials (~18%), I would have expected

their performance on the Morris watermaze to be more variable – sometimes performing similar to the WT mice and sometimes performing significantly worse.

8. There are several points in the paper where the language used is confusing. For example - I am not sure what is meant by 'multi-stability', 'long timescale multi-stability' or the term 'episodic spatial map instability'.

9. Why did the cue card have a bottle affixed to it?

10. The statement that this mouse is a manipulation of "synaptic and intrinsic potentiation" is perhaps too oversimplified. More details regarding this specific knockout mouse should be provided.

11. Use of k-means algorithms require a priori justification for the number of clusters enforced on the data or post-hoc analysis to determine that the number of clusters was correct. I could not locate information to indicate that the authors took these considerations into account?

Reviewer #2 (Remarks to the Author):

This paper presents novel data showing a decrease in the stability of place cell representations in mice in which long-term depression is deficient at the parallel fiber-Purkinje cell synapse. Overall, the results are convincing and clearly presented for the most part. However, there are some issues with the manuscript that should be rectified. My specific comments are as follows:

1. The importance of the results would be clearer if the authors thoroughly explained the anatomical pathways by which information coded by the Purkinje cells reaches the hippocampus (e.g., this could perhaps be explained in the second paragraph of the Introduction).
2. The authors should report statistics and degrees of freedom, not just p-values. Also, the authors should be sure to present statistics in the text for negative results, not just positive results.
3. It was somewhat surprising to me that intra-session stability was normal while inter-session stability was impaired. The authors should perhaps explain more clearly what this suggests.
4. I was also somewhat surprised that spatial information was unaffected in the mutant mice. I would have expected spatial information to decrease in conjunction with increases in field size and decreases in spatial coherence. The authors should perhaps clearly explain what this implies or why this would be the case. Is it possible that the spatial information measure is simply less sensitive than the other measures?
5. In Figure 1A, top right panels, the authors show an example control place cell that remaps between S1 and S2, but this was not common. Therefore, the authors should probably point this out in the figure legend.
6. In Figure 2B-C, it appears the authors did not include mutant and control data in the same statistical analysis. Yet, the authors seem to use these analyses to imply that mutant and control animals are different in this measure. However, they are missing the interaction effect (i.e., between spatial map stability categorization and genotype) that is necessary to support this conclusion (See Nieuwenhuis et al., Nature Neuroscience, 2011). A similar point also applies to the data shown in panels D-E and F-G.
7. Related to the above point, it is unclear what statistical test was performed to conclude that there was no effect of darkness on field sizes in control and mutant mice. Again, the authors should have included both light and darkness and both genotypes in the same analysis, shown no significant interaction effect, no significant main effect of light condition, and a significant main effect of genotype to support their conclusions.
8. The example place cells in Figure 3G show that the fields in the S5 session revert to their locations from S1. Did this always happen or was this just a coincidence in this example?

9. The statistical result that is reported in the Figure 4D legend is ambiguous. Does this indicate a main effect of training or genotype?
10. Why were behavioral tests conducted during the light phase? Isn't that when the mice were sleeping?
11. In Figure S1B, what is the difference between trials 1-4 on the left and trials 1-4 on the right?
12. In the example trajectories in Figure S3, it appears L7-PP2B mice avoided the center of the arena compared to control mice. Was that true across all mice or just an anomaly in this example? If this was a consistent observation across different animals, it should be quantified and discussed.

Reviewer #3 (Remarks to the Author):

SUMMARY:

The experiments examine hippocampal place cell activity and spatial learning in L7-PP2B mice with impaired LTP in Purkinje cells. The authors make three claims based on their results: (1) The place cell representation of L7-PP2B is "unstable" (i.e. place fields form when the mouse is introduced in an arena containing a single proximal cue, but the place fields often rotate if the mouse is taken out of the arena and reintroduced again 4 minutes later). (2) This instability of hippocampal place cells results from an incorrect anchoring of the spatial map on the visual cue (allocentric representation), NOT from an incorrect anchoring on self-motion cues (body-centered representation). (3) The impaired allocentric representation causes spatial disorientation in the L7-PP2B mice as measured by impaired performance in a free exploration task and the Morris watermaze.

SIGNIFICANCE:

Previous work by the same authors has already demonstrated that cerebellar plasticity (Purkinje cell-LTD) is critical for generating spatial representations in the hippocampus, when these spatial representations rely on self-motion cues. What is significant is that the current findings point to a different type of interaction, where other types of cerebellar plasticity (Purkinje cell-LTP) may play a role in spatial navigation by helping the hippocampus to generate stable spatial representations that rely on allocentric visual cues. Whether the current study will have a major impact on the field will depend on clearly demonstrating this link.

MAJOR POINTS TO ADDRESS:

1) The authors have previously used a number of task conditions to dissociate deficits in allocentric vs. self-motion spatial representations. In contrast, the current paper only uses a cue removal condition to show that place cell properties of L7-PP2B mice stay the same after removing the visual cue and turning off the lights. While the authors interpret this result to claim that L7-PP2B mice are normal with respect to spatial navigation based on self-motion cues, this conclusion needs to be strengthened with additional tests. In their previous work, the authors checked watermaze performance in the dark to show that mice deficient in Purkinje cell-LTD are impaired in using self-motion cues to find the hidden platform. Perhaps a similar test can be used here to evaluate performance during self motion-based navigation in L7-PP2B mice.

2) The major claim in the manuscript is that L7-PP2B mice have high susceptibility to spatial map instability, and that this instability leads to spatial disorientation. However, some of the results are difficult to reconcile with this conclusion:

First, Figure 2B,D,F demonstrate that performance in the free exploration task is normal in a wildtype mouse with unstable hippocampal spatial map representations. Based on this result, there is no reason to suspect that similar unstable representations could cause any of the behavioral impairments reported for L7-PP2B mice in the free exploration task (Figure 2C,E,G).

Second, the impairments in watermaze performance of L7-PP2B mice are very subtle. In fact, L7-

PP2B mice learn to find the hidden platform by swimming the same distance as controls (Figure 4A), and the deficits reported only show up as small differences in the initial segment of their swim trajectory (Figure 4B-E). It is peculiar that these subtle deficits are present from the first session of training (S1) presumably before learning has taken place. These effects may be more consistent with a subtle deficit in motor performance than a deficit caused by spatial disorientation. Given that L7-PP2B mice have impairments in basic motor function of vestibular reflexes (Schonewille et al., Neuron 2010), it is critical that the authors determine the root cause of the subtle behavioral impairments reported in the watermaze task.

For example, since the authors claim that impaired performance in the watermaze is the result of unstable hippocampal maps that rotate between different exposures and cause disorientation, it would be useful to compare the watermaze performance of L7-PP2B mice with two other control groups: (1) normal wildtype mice that are disoriented before each training session and (presumably) have unstable place field representations, (2) normal wildtype mice that are trained in an unstable environment that "rotates" from session to session (or alternatively, an unstable environment in which the hidden platform moves from session to session according to a fixed random rotation).

Lastly, it is critical that the authors evaluate whether the place fields of L7-PP2B mice are unstable under conditions that more closely resemble what the mice experience in the watermaze task (which include multiple prominent distal cues and are very different from the conditions experienced during free exploration in a stark arena with a single proximal cue). Without this kind of information it is impossible to make any claims about what may be causing the subtle deficits of L7-PP2B mice in the watermaze task, or whether their place cell representations are unstable in this task at all.

MINOR POINTS:

- 1) Specify whether experiments were run blind as to genotype. If not, what kind of precautions were taken to make sure that all mice were handled in exactly the same way? This is particularly critical for the place cell experiments, given that the phenotype reported is a place field rotation that can occur in both wildtype and L7-PP2B mice.
- 2) Specify how many S1-S2 sessions were experienced by each mouse. If more than one, was the location of the cue changed between one S1-S2 session and the next S1-S2 session?
- 3) Use same format as in Figure 3I (right panel) to specify number of cells for each mouse in Figure 1D,E.
- 4) Figure 2C indicates that place cells were "Always stable" in 4 out of 7 of the L7-PP2B mice. A statistical analysis is necessary to determine whether this is a consequence of the probabilistic (sporadic) nature of the place field rotations, or whether the phenotype is absent in a population of L7-PP2B mice. If the place cell instability phenotype is really absent in many of the L7-PP2B mice, it will be critical to evaluate performance in the watermaze separately for L7-PP2B mice with and without the phenotype.
- 5) Specify the results in Figure 2 separately for S1 and S2, not as a ratio S1/S2. In addition to the bars showing the average value and the standard deviation in each one of the plots, show the individual points that comprise the average.

Reviewers' comments:

Reviewer #1 (Remarks to the Author):

In this manuscript, the authors used L7-PP2B knockout mice to examine how the loss of long-term potentiation and intrinsic excitability impact hippocampal spatial presentations. They report decreased stability of place cells in a cued circular environment in L7-PP2B mice compared to control mice. This instability was bi-modal; all place cells in a session either remained stable or remapped between S1 and S2. Unstable sessions were associated with increased exploration of the cue. Visual cue removal did not affect stability but reintroducing the mice to the environment again caused instability. Lastly – the L7-PPB mice exhibit deficits in the self-localization abilities in the spatial version of the Morris Water Maze. Dissecting the role of the cerebellum in generating stable neural representations of space is an important topic and the author's use of a unique knockout animal to target a specific feature of cerebellar processing is novel and interesting. However, I have serious concerns about the degree to which the phenotype they observed is robust, the level of clarity regarding the potential mechanism underlying their observations and how the behavioral data fits with the effects they observed in the hippocampal neural data.

We thank the reviewer for the constructive comments on the manuscript. All reviewer's concerns have been addressed as described below.

1. Generally speaking, I was not convinced by the electrophysiological effects presented. The knockout of PP2B in Purkinje cells appears to marginally increase the amount of sporadic place cell remapping that occurs across behavioral sessions (3/53 sessions in wildtype compared to 11/61 sessions in knockout mice). The numbers of 'unstable sessions', as well as the number of knockout mice that compose these sessions (n = 3), are both very low.

We acknowledge that the number of recorded mice and unstable sessions was low. We therefore performed more electrophysiological recording sessions from both control and mutant mice. The manuscript now presents 57 sessions from 8 controls and 73 sessions from 8 mutant mice. In addition, we re-analyzed datasets in order to improve quantification of place cells instability. Indeed, we need to stress here that the quantification we used previously was too drastic and artificially underestimated the phenotype in two ways.

1/ Unstable cells coming from sessions in which only one place cell was recorded were not included in the pool of unstable sessions but were taken into account for calculating the total number of sessions. This reduced artificially the proportion of unstable sessions (Fig. 1 F) as well as the number of sessions considered for the behavioral correlate (Fig. 2). These sessions are now considered and included in the analyses of map instability (Fig. 1 and Fig. 2). The reviewer will find examples of such sessions in the figure presented below.

Examples of unstable cells coming from sessions in which only one place cell was recorded

2/ The apparent proportion of mutant mice displaying instability was misleading (former Fig. 2). Indeed, considering that occurrence of instability is ~ 25 % (see revised Fig. 1), a mouse can only be classified as stable or unstable if more than 4 recording sessions have been collected. Thus, out of the 6 L7-PP2B mice reaching this criterion, 5 showed instability.

Improving accuracy of the analysis using these criteria better unveils the phenotype of L7PP2B mice, now showing 25% of unstable sessions (18/73, observed in 5 mice) against 5% in control mice (3/57, observed in only 1 mouse). We have implemented the new version of the manuscript with these results and changed the figure accordingly (Fig. 1).

2. In addition to the low n, I'm not sure how to interpret the bi-modal effect the authors observe on place cell stability. If there is a significant deficit in the ability of these animals to maintain a stable allocentric reference frame, why was the change in stability observed in so few sessions? Instead, this suggests to me, there could be some other underlying effect causing the sporadic remapping. For example, in Figure 2, it appears that coverage of the environment was not as uniform in sessions that were picked up as being unstable. A systematic coverage distance would not be picked up by just calculating traveled distance (as the mice could travel a very long distance but never enter the center of the arena). If the coverage is low in the center of the box, it could be that place cells appear to shift simply because the mouse failed to uniformly sample the entire environment and thus, aberrant spikes were defined incorrectly as place fields. Or perhaps the animals were placed in the environment from very different positions or angles on certain S1 to S2 trials compared to other trials.

As mentioned in response to point #1, correcting the quantification of place cells instability with more accurate criteria revealed that instability occurred in 25 % of the sessions (Fig. 1 F). Importantly, instability does not correspond to sporadic remapping but to an unstable orientation of the spatial map, since during shift events place cells relationships remain coherent but rotate relative to the landmark cue. Other investigations on rats intentionally disoriented before each session of exploration (Knierim et al., 1995) or on old rats impaired for hippocampal plasticity (Barnes et al., 1997) have also reported spatial map instability occurring only in few sessions. In both studies spatial map shift occurred in ~30 % of the sessions, corresponding to the same range as our results and thus suggesting that spatial map instability is a discontinuous phenomenon.

The reviewer expressed concerns regarding a possible inaccuracy in defining place field due to aberrant spikes. We would like to point out that Fig. 1A, Fig. S5 as well as new cells added to the dataset (point #1), display raw maps showing spikes in red and animal's path in black, thus illustrating the shift in spike location that occurs during remapping. These maps clearly exemplify that spatial map rotation is not due to artefactual field detection. In addition, the reviewer will find below two other examples of raw maps from unstable sessions.

Two additional examples of raw maps from unstable sessions

As mentioned by the reviewer, the impression of a non-uniform coverage comes from figure 2. This is because this figure focuses on the 2 first minutes of exploration to illustrate the changes in mouse behavior when their map shifted, i.e. an increased exploration of the object. Importantly these tracks do not represent the overall coverage of the arena corresponding to the 12 min of exploration. To make this clear, we have modified figure 2, which now mention the 2 min period.

Nevertheless, as suggested by the reviewer, we assessed mice exploratory behavior in the center versus periphery of the arena by quantifying the distance to the wall, the percentage of time spent in the center as well as the number of entries in the center. None of these parameters were different between controls and L7-PP2B mice (see detailed statistics below). In view of these analyses, we conclude that the difference in the occurrence of instabilities between genotypes is not attributable to a differential exploration of the center.

Exploratory behavior in the center vs periphery of the arena is normal in L7-PP2B mice.

Statistical comparison between controls and L7-PP2B mice:

- mean distance from wall (normal distribution): genotype effect, $F_{(1,133)}=0.93$, $p=0.34$; genotype*session effect, $F_{(1,133)}=0.23$, $p=0.63$, repeated measure ANOVA with two factors,
- % of time in the center (data are not normally distributed) : Session 1, $U=1834$, $p=0.124$; session 2, $U=1962$, $p=0.386$; Mann-Whitney U-test with p values adjusted for multiple comparisons.
- Normalized number of entries in the center (normal distribution): genotype effect $F_{(1,133)}=1.01$, $p=0.32$; genotype*session effect $F_{(1,133)}=0.79$, $p=0.38$, repeated measure ANOVA with two factors

Finally, the reviewer expressed concerns regarding the way mice were placed in the environment. We would like to specify that for each recording session, mice were systematically placed at the same position in the arena, in front of the object. Hence, phenotype of instability could thus not result from animals being “*placed in the environment from very different positions or angles on certain S1 to S2 trials compared to other trials.*” This is now mentioned in the method part of the revised manuscript (page 3, lines 20-21).

3. The take-home message of the paper is also not clear to me. The authors report a bi-modal effect, where knockout mice had slightly more sessions in which place cells coherently remapped. However, how this effect relates to LTP in Purkinje cells, as well as how to relate this effect to a difference in short versus long path lengths in the Morris Water Maze is not clear to me. The mechanism underlying the place cell instability is also not clear. One potential way the paper could be improved would be to add simultaneous recordings of head direction cells. The presence of drift or remapping in the head direction cell population might at least point to a potential interpretation of the effects reported in place cells.

The take home message of our manuscript is that PP2B-dependent Purkinje cell (PC) potentiation is required for both, a stable orientation of the place code in an allocentric reference frame and accurate spatial navigation. Here we did not intend to causally link a deficit in the place code with impaired Water Maze performance. Instead, we questioned how a deficit in PP2B-dependent PC potentiation impacts these two processes. Interestingly, we found that L7-PP2B mice impairments in the Morris Water Maze, illustrated by an increase in the variability of their performances (Fig. 5C) and inaccurate initial orientation (Fig. 5D) are compatible with an unstable orientation of spatial representation.

The question of how place cell instability relates to LTP in Purkinje cells is indeed important. Impaired cerebellar PC activity can impact hippocampal activity through two major pathways: the vestibular nuclei-HD system on the one hand (see for example Smith, 2005) as suggested by the referee and the deep cerebellar nuclei-parietal cortex pathway on the other hand (Giannetti and Molinari, 2002). Exploring the impact of one or the other of these different pathways on hippocampal activity represents in itself a whole new investigation that will be pursued in future dedicated studies.

To address the point raised by the reviewer about the mechanisms underlying spatial map instability, we investigated hippocampal local field potential (LFP) activity in control and L7-PP2B mice during two consecutive sessions of free exploration of the familiar environment (new Fig. 4). LFP activity in theta (6-12 Hz) and low-gamma (20-45 Hz) bands was not different between controls and L7-PP2B mice (Fig 4B-C, *theta*: genotype, $F_{(1,8)} = 1.24$, $p=0.30$; session, $F_{(1,8)} = 2.44$, $p=0.16$; session * genotype interaction $F_{(1,8)} = 0.31$, $p=0.59$; *low gamma*: genotype, $F_{(1,8)} = 1.96$, $p=0.20$; session, $F_{(1,8)} = 3.32$, $p=0.1$; session * genotype interaction $F_{(1,8)} = 3.70$, $p=0.09$, ANOVA). However an increase in the high gamma band was observed in L7-PP2B mice compared to control mice in session S2 specifically (Fig 4B-C, genotype, $F_{(1,8)} = 5.3$, $p=0.05$; session, $F_{(1,8)} = 6.6$, $p=0.03$; session * genotype interaction, $F_{(1,8)} = 6.1$, $p=0.04$; L7-PP2B, S1-S2, $p= 0.007$, controls-L7-PP2B in S2, $p=0.007$, LSD post-hoc test). This result points towards an increase in the computation linked with high gamma in the CA1 region of L7-PP2B mice specifically when mice re-enters the familiar environment. Gamma rhythms have been proposed to modulate interactions between substructures of the hippocampal formation such that low- and high-frequency gamma oscillations would mediate the coherence between CA1 and CA3 or entorhinal cortex (EC), respectively (Bragin et al., 1995; Colgin et al., 2009). Thus, the low gamma/high gamma ratio in CA1 may reflect its input dominance toward a memory-based/CA3 versus a contextual/EC computational mode (Cabral et al., 2014). Here, the increased high gamma computation in L7-PP2B points toward an unbalance in favor of EC inputs towards CA1 that may induce a bias in the integration

of external sensory inputs, eventually leading to unstable anchoring of the spatial map to the allocentric frame. We have now added these new analyses and included the latter discussion in the revised manuscript (Fig. 4 and page 14-15).

4. For Figure 2 – the increased exploration of the cue in sessions where the rotation occurs could indicate that the animals think the object has been displaced (or is novel). If you use the same environment but rotate the cue – do KO mice explore the cue more than WT mice?

Indeed, we agree with the reviewer that the increased exploration during unstable sessions might indicate that the mouse considers the object as novel or displaced. As suggested by the reviewer, we have performed experiments in which we rotated the cue in the same environment and analyzed the consequence on control and L7-PP2B behavior. Since the environment is symmetrical relative to the cue, the cue rotation was performed in front of the animal in order to be detected. Thus, the animal was free to explore the familiar environment for a 12 min session (“familiar position” session) and then the cue was rotated in the presence of the animal. Cue rotation was followed by a 12 min session of free exploration with the cue displaced (“new position” session). Exploratory behavior was analyzed during the 2 first minutes of the new position session and compared to the 2 first minutes of the familiar position session. We found that object rotation induced a clear increase of the global exploration of the arena as evidenced by the increase of the travelled distance and the decrease in the number of stops for both genotypes (see following figure, $p_{\text{session}} < 0.001$ for both parameters, ANOVA). Moreover, Control and L7-PP2B mice displayed a more pronounced exploration of the object, shown by the decrease of the mean distance from the cue and an increase in the number of entries in the cue zone (see following figure, $p_{\text{session}} = 0.04$ and $p_{\text{session}} < 0.001$ respectively, ANOVA). Importantly, both groups of mice re-explored the arena and the cue the same way after cue rotation (ANOVA, $p_{\text{genotype}} > 0.05$ for all parameters). Besides, for all behavioral parameters, no interaction was found between sessions and genotypes (ANOVA, all $p_{\text{interaction}} > 0.05$).

Visible cue rotation induces a similar increase in control and L7-PP2B exploratory behavior. (A) Examples of control and L7-PP2B trajectories before (familiar position) and after (new position) a visible displacement of the object. (B-E) bar plots showing the total travelled distance (B), the number of stops (C) the distance from the cue (D) and the number of entries in the cue zone (E) before and after cue rotation. These data illustrate that both genotype displayed an increase in the travelled distance (B, session, $F_{(1,43)} = 46.05$, $p < 0.001$; genotype, $F_{(1,43)} = 2.67$, $p=0.11$; session * genotype interaction, $F_{(1,43)} = 0.57$, $p=0.45$), a decrease in the number of stops (C, session, $F_{(1,43)} = 16.16$, $p < 0.001$; genotype, $F_{(1,43)} = 2.46$, $p=0.12$; session * genotype interaction, $F_{(1,43)} = 0.17$, $p=0.68$), a decrease in the distance from the cue (D, session, $F_{(1,43)} = 4.5$, $p = 0.040$; genotype, $F_{(1,43)} = 0.803$, $p=0.38$; session*genotype interaction, $F_{(1,43)} = 0.29$, $p=0.59$) and an increase in the number of entries in the cue zone (E, session, , $F_{(1,43)} = 28.9$, $p < 0.001$; genotype, , $F_{(1,43)} = 2.02$, $p = 0.16$; session*genotype interaction, $F_{(1,43)} = 0.15$, $p=0.70$) when the cue has moved to a new place (two-way ANOVA with repeated measures).

In conclusion, these new data indicate that both control and mutant mice respond to object displacement by increasing exploration of the arena and of the displaced object. Remarkably, this supports our interpretation that the object is perceived as displaced during the directional shift of the hippocampal spatial map. We have now incorporated these data in the revised manuscript (page 11-12 and Fig. S7).

5. Can the authors dissociate the impact of the change in place cell field size versus stability on the behavioral deficits they observed in knockout animals?

In the Morris Water Maze, we found that L7-PP2B orientation behavior is impaired as revealed by analyses of their position soon after they started to swim (Fig. 5D) as well as the location of their research area (Fig. 5E). Our additional analysis indicated that their performances are also more variable (see answer to comment #6 and Fig. 5C). Together these findings are coherent with an impaired spatial map stability rather than an increased field size. However, since we cannot directly link the deficit observed in the Morris Water Maze with the type of alteration found in the hippocampal spatial code we decided to be more cautious in our interpretation and have modified the text accordingly (see page 19 lines 5-7 and abstract).

6. Given that L7-PP2B mice show instability only in some trials (~18%), I would have expected their performance on the Morris watermaze to be more variable – sometimes performing similar to the WT mice and sometimes performing significantly worse.

We thank the reviewer for this suggestion. Variability in mouse's performances was inspected by analyzing the number of transition between successful and failed trials. Failed trials were identified using a gaussian fit on the distribution of travelled distances (see methods). Since the gaussian fit was centered on the successful trial distances, the threshold was defined as the distance above which the fit could not explain more than 90% of the distribution (note that a bimodal gaussian fit could not be used since the distribution of failed trials could not be fitted by a gaussian curve). Using the threshold defined above (678 cm), we found that L7-PP2B mice performances were indeed more variable as illustrated by the increase in successful/failed trials transitions during the last 2 days of training (i.e. when the performances of control mice reached a plateau) ($p=0.012$, Fig. 5C). This new result is now added in the revised manuscript (page 18, lines 8-12).

7. There are several points in the paper where the language used is confusing. For example - I am not sure what is meant by 'multi-stability', 'long timescale multi-stability' or the term 'episodic'

We improved the language to avoid any confusion.

8. Why did the cue card have a bottle affixed to it?

In our study, mice are not food deprived and the exploration of the environment relies exclusively on spontaneous behavior. The use of a 3D object increases their motivation to explore.

9. The statement that this mouse is a manipulation of "synaptic and intrinsic potentiation" is perhaps too oversimplified. More details regarding this specific knockout mouse should be provided.

We have now added precisions regarding the phenotype of the L7-PP2B mouse in the appropriate method section of the revised manuscript (page 1 lines 12-19 of the methods).

10. Use of k-means algorithms require a priori justification for the number of clusters enforced on the data or post-hoc analysis to determine that the number of clusters was correct. I could not locate information to indicate that the authors took these considerations into account?

We thank the reviewer for this remark. In the revised manuscript, the K-mean algorithm is only used for the separation of the stable cell and unstable cell clusters. In order to assess the relevance of our number of clusters, we used the Elbow method, which consists in choosing the number of clusters so that adding one more cluster only slightly improves modeling of the data. In a plot representing the within variance group, it corresponds to the angle of the graph, i.e., in our data presented below, 2 clusters. This has been added in the methods (page 16, lines 16-18).

Reviewer #2 (Remarks to the Author):

This paper presents novel data showing a decrease in the stability of place cell representations in mice in which long-term depression is deficient at the parallel fiber-Purkinje cell synapse. Overall, the results are convincing and clearly presented for the most part. However, there are some issues with the manuscript that should be rectified. My specific comments are as follows:

We thank the reviewer for the constructive comments on the manuscript. All concerns of the reviewer have been addressed as described below.

1. The importance of the results would be clearer if the authors thoroughly explained the anatomical pathways by which information coded by the Purkinje cells reaches the hippocampus (e.g., this could perhaps be explained in the second paragraph of the Introduction).

We now have included this information in the second paragraph of the introduction (page 3-4).

2. The authors should report statistics and degrees of freedom, not just p-values. Also, the authors should be sure to present statistics in the text for negative results, not just positive results.

We agree with the reviewer and have now systematically reported statistics and degrees of freedom for each p-value indicated in the text or in the legends of the figures. Additionally we now report in the revised version of the manuscript statistics for negative results.

3. It was somewhat surprising to me that intra-session stability was normal while inter-session stability was impaired. The authors should perhaps explain more clearly what this suggests.

Intra-session stability measures the ability of a firing map to remain stable during a given period. This analysis compares the firing map from the first minutes of a session with the last minutes of the same session. Importantly this comparison takes place between periods during which the animal is not removed from the arena. Remarkably, the fact that intra-session stability is normal in L7-PP2B mice while inter-session stability is altered actually suggests that the ability of the spatial map to correctly anchor to a landmark is altered specifically when the animal has first to reset its initial orientation according to an external landmark (i.e. when it is removed from and placed back into the arena), but once the orientation is settled, it remains stable as long as the animal stays in the place.

As suggested by the reviewer, we now discuss this point in the revised manuscript (p.9, lines 1-4).

4. I was also somewhat surprised that spatial information was unaffected in the mutant mice. I would have expected spatial information to decrease in conjunction with increases in field size and decreases in spatial coherence. The authors should perhaps clearly explain what this implies or why this would be the case. Is it possible that the spatial information measure is simply less sensitive than the other measures?

Spatial information content (in bits per spike) is calculated using the general formula of information content: $I = \sum_i \lambda_i (A_i/A) \times \log_2 (A_i/A) \times P_i$ where A_i is the mean firing rate in each pixel, λ is the overall mean firing rate, and P_i is the probability of the animal to be in pixel i (i.e. occupancy in the i -th bin / total recording time). Spatial information content is a refined measure that relies on the contrast of firing rate between pixels of the map. It can be approximated by the infield rate/outfield rate ratio. Regarding our data, we indicated in the manuscript that spatial information was not significantly affected in mutant mice compared to control mice. However, it is important to point out that the exact p value is $p = 0.066$ (Mann-Whitney U-test). The absence of difference should thus be interpreted with caution. The exact p-value is now indicated in figure S3.

5. In Figure 1A, top right panels, the authors show an example control place cell that remaps between S1 and S2, but this was not common. Therefore, the authors should probably point this out in the figure legend.

We thank the reviewer for this suggestion and have modified the legend accordingly.

6. In Figure 2B-C, it appears the authors did not include mutant and control data in the same statistical analysis. Yet, the authors seem to use these analyses to imply that mutant and control animals are different in this measure. However, they are missing the interaction effect (i.e., between spatial map stability categorization and genotype) that is necessary to support this conclusion (See Nieuwenhuis et al., Nature Neuroscience, 2011). A similar point also applies to the data shown in panels D-E and F-G.

We agree with the reviewer that comparison of control and mutant data should be done by including their data in the same analysis. We did not aim at implying that mutant mice are different from controls in their cue exploratory behavior during spatial map rotation. Yet, in the control mice, spontaneous rotations of the spatial map was sporadic and only observed during 3 sessions in one mouse. Which such a low statistical power, we can't reliably test the interaction effect between control and L7-PP2B mice, and even reliably interpret in term of behavior the dataset from the controls. Since the question addressed in the figure 2 is whether spatial map rotation is associated with a modification in mouse behavior we compared stable and unstable sessions inside each genotype, without measuring the genotype effect.

We have now improved the figure by presenting scatter plots of all datapoints which allows for a more direct reading of the datasets and S1 and S2 separately (following reviewer 3' suggestions), in addition to the ratios. Statistical analysis using non parametric Wilcoxon Signed-Ranks test and adjusted p values for multiple comparisons revealed an increase in object exploratory behavior in S2 relative to S1 during sessions with spatial map rotation in L7-PP2B mice, whereas this was not the case during stable sessions (mean distance from the cue: stable sessions $Z=0.82$, $p=0.82$; spatial map rotation, $Z=3.24$, $p=0.0024$; % of time in the cue zone: stable sessions $Z=0.20$, $p=1$; spatial map rotation, $Z=2.37$, $p=0.0368$; normalized number of entries in the cue zone: stable sessions $Z=0.017$, $p=1$; spatial map rotation, $Z=2.77$, $p=0.011$). Although no difference was found in object behavior during spatial map rotation in the control group (mean distance from the cue: stable sessions $Z=1.59$, $p=0.22$; spatial map rotation, $Z=1.07$, $p=0.58$; % of time in the cue zone: stable sessions $Z=1.47$, $p=0.284$; spatial map rotation, $Z=1.60$, $p=0.22$; normalized number of entries in the cue zone: stable sessions $Z=2.38$, $p=0.035$; spatial map rotation, $Z=1.60$, $p=0.221$), this lack of significance should be interpreted with caution and might simply result from the low sample size. Indeed, the graph shows a clear tendency of the control as well to explore more the object during unstable sessions compared to stable ones. This analysis therefore supports our conclusions that spatial map instability leads to disorientation.

7. Related to the above point, it is unclear what statistical test was performed to conclude that there was no effect of darkness on field sizes in control and mutant mice. Again, the authors should have included both light and darkness and both genotypes in the same analysis, shown no significant interaction effect, no significant main effect of light condition, and a significant main effect of genotype to support their conclusions.

We have now performed statistical analyses as asked by the reviewer. Effect of darkness on field size was analyzed using an ANOVA, which revealed a highly significant genotype effect ($p<0.001$), no significant main effect of light vs dark condition ($p=0.42$) and a marginal although not significant genotype*session interaction effect ($p=0.058$), importantly, no difference was found in L7-PP2B mice between light and dark conditions ($p=0.46$, LSD post-hoc test). The values are now implemented in the revised version of the manuscript (legend Fig. 3).

The same analysis has now been performed on all other parameters of the figure (spatial coherence, spatial information and intra-session stability) and showed no significant effect.

8. The example place cells in Figure 3G show that the fields in the S5 session revert to their locations from S1. Did this always happen or was this just a coincidence in this example?

We observed both situations: the fields went back to S1 position in 75 % of the sessions, and were located at a new position in 25% of the cases. We have now indicated these proportions in the revised manuscript (p. 14, lines 14-15).

9. The statistical result that is reported in the Figure 4D legend is ambiguous. Does this indicate a main effect of training or genotype?

It indicates a main effect of genotype. We have now added all statistics in the legend.

10. Why were behavioral tests conducted during the light phase? Isn't that when the mice were sleeping?

Although mice and rats are more active in the dark than during the light phase, several investigations have shown that the acquisition of the Morris Water Maze is not affected by the day period in those species (Aslani et al., 2014, Martin-Fairey and Nunez, 2014).

11. In Figure S1B, what is the difference between trials 1-4 on the left and trials 1-4 on the right?

The left and right parts correspond to different rotarod speeds (5 and 10 rounds per min, respectively). We thank the reviewer for pointing out this omission and have now rectified the figure to indicate speeds on the graph.

12. In the example trajectories in Figure S3, it appears L7-PP2B mice avoided the center of the arena compared to control mice. Was that true across all mice or just an anomaly in this example? If this was a consistent observation across different animals, it should be quantified and discussed.

To address this question, we have investigated several parameters relating to mouse exploratory behavior in the center of the arena in both session 1 and session 2. We analyzed the distance to the wall, the percentage of time spent in the center as well as the number of entries in the center and found none of these parameters to be different between controls and L7-PP2B groups (see detailed statistics below). Therefore, the observation pointed out by the reviewer is not consistent across different animals and is a particularity of the example shown in Figure S3. We therefore added additional examples to avoid misinterpretation of the global L7-PP2B behavior (Fig.S3).

Exploratory behavior in the center vs periphery of the arena is normal in L7-PP2B mice.

Statistical comparison between controls and L7-PP2B mice :

-mean distance from wall (normal distribution): genotype effect, $F_{(1,133)}=0.93$, $p=0.34$; genotype*session effect, $F_{(1,133)}=0.23$, $p=0.63$, repeated measure ANOVA with two factors,

- % of time in the center (data are not normally distributed) : Session 1, $U_{(1,133)}=1834$, $p=0.124$; session 2, $U_{(1,133)}=1962$, $p=0.386$; Mann-Whitney U-test, with p values adjusted for multiple comparisons.

-Normalized number of entries in the center (normal distribution) : genotype effect $F_{(1,133)}=1.01$, $p=0.32$; genotype*session effect $F_{(1,133)}=0.79$, $p=0.38$, repeated measure ANOVA with two factors.

Reviewer #3 (Remarks to the Author):

SUMMARY:

The experiments examine hippocampal place cell activity and spatial learning in L7-PP2B mice with impaired LTP in Purkinje cells. The authors make three claims based on their results: (1) The place cell representation of L7-PP2B is “unstable” (i.e. place fields form when the mouse is introduced in an arena containing a single proximal cue, but the place fields often rotate if the mouse is taken out of the arena and reintroduced again 4 minutes later). (2) This instability of hippocampal place cells results from an incorrect anchoring of the spatial map on the visual cue (allocentric representation), NOT from an incorrect anchoring on self-motion cues (body-centered representation). (3) The impaired allocentric representation causes spatial disorientation in the L7-PP2B mice as measured by impaired performance in a free exploration task and the Morris watermaze.

We thank the referee for the summary of our results. In particular, we agree with points (1), (2) and the part of point (3) concerning the link between the impaired allocentric representation and the spatial disorientation in a free exploration task. However, our intent with the water maze task was not to causally link impaired place cells representation with the behavioral disorientation observed in this task but rather to explore how PP2B-dependent mechanisms may impact spatial orientation during goal-directed navigation. Interestingly, we found that L7-PP2B mice impairments in the Morris Water Maze, illustrated by an increase in the variability of their performances (Fig. 5C) and inaccurate initial orientation (Fig. 5D) are compatible with an unstable orientation of spatial representation. Nevertheless, the take home message of our manuscript is that PP2B-dependent Purkinje cell (PC) potentiation is crucial for both, a stable orientation of the place code in an allocentric reference frame and for accurate spatial navigation.

Additionally, our conclusion about the “instability of hippocampal place cells that results from an incorrect anchoring of the spatial map on the visual cue (allocentric representation), NOT from an incorrect anchoring on self-motion cues (body-centered representation)” is now reinforced by new experiments and analyses (see answer to comment 1 below)

SIGNIFICANCE:

Previous work by the same authors has already demonstrated that cerebellar plasticity (Purkinje cell-LTD) is critical for generating spatial representations in the hippocampus, when these spatial representations rely on self-motion cues. What is significant is that the current findings point to a different type of interaction, where other types of cerebellar plasticity (Purkinje cell-LTP) may play a role in spatial navigation by helping the hippocampus to generate stable spatial representations that rely on allocentric visual cues. Whether the current study will have a major impact on the field will depend on clearly demonstrating this link.

MAJOR POINTS TO ADDRESS:

1) The authors have previously used a number of task conditions to dissociate deficits in allocentric vs. self-motion spatial representations. In contrast, the current paper only uses a cue removal condition to show that place cell properties of L7-PP2B mice stay the same after removing the visual cue and turning off the lights. While the authors interpret this result to claim that L7-PP2B mice are normal with respect to spatial navigation based on self-motion cues, this conclusion needs to be strengthened with additional tests. In their previous work, the authors checked watermaze performance in the dark to show that mice deficient in Purkinje cell-LTD are impaired in using self-

motion cues to find the hidden platform. Perhaps a similar test can be used here to evaluate performance during self motion-based navigation in L7-PP2B mice.

We agree with the reviewer that the importance of our findings also depends on clearly demonstrating that self-motion based spatial representation, free exploration and navigation of L7-PP2B mice is unaffected.

L7-PP2B mice display normal spatial representation when exploring the arena using self-motion (in the dark, and without proximal cue, Fig. 3). In such dark conditions free exploration of L7-PP2B mice is also intact. We indeed analyzed general locomotor activity (travelled distance, number of stops and speed) as well as exploratory behavior (distance to the wall, % of time spent in the center and number of entries in the center) and found no difference between control and L7-PP2B mice (all $p > 0.05$, Mann Whitney U-test see below). These data indicate that self-motion based exploration as well as self-motion based representation of a familiar place is unaffected. We have now included these new results in the revised manuscript (Fig. S8).

Exploratory behavior in self-motion dependent context (in the dark, without the object) is unaltered in L7-PP2B mice. A-D), scatter plots showing that control and L7-PP2B mice display no difference in their general locomotor activity in the dark, as illustrated by the traveled distance (A, $U=186$, $p=0.76$), the number of stops (B, $U=196$, $p=0.96$) or the mean speed (C, $U=171$, $p=0.47$). The center versus periphery exploratory behavior was also similar between control and mutant mice. No difference was found in the mean distance from the wall (D, $U=173$, $p=0.51$), the percentage of time spent in the center (E, $U=150$, $p=0.20$) or the normalized number of entries in the center (F, $U=150$, $p=0.20$). Mann-Whitney U-test.

Finally, as requested by the reviewer, we also tested the abilities of L7-PPB to navigate toward a goal using self-motion information. Using a self-motion based version of the Morris Water Maze, we

found no difference between control and L7-PP2B mice navigation performances (see figure below). Our data thus confirm that L7-PP2B abilities to navigate using self-motion information are unaffected.

Self-motion based navigation is unaltered in L7-PP2B mice

2) The major claim in the manuscript is that L7-PP2B mice have high susceptibility to spatial map instability, and that this instability leads to spatial disorientation. However, some of the results are difficult to reconcile with this conclusion: First, Figure 2B,D,F demonstrate that performance in the free exploration task is normal in a wildtype mouse with unstable hippocampal spatial map representations. Based on this result, there is no reason to suspect that similar unstable representations could cause any of the behavioral impairments reported for L7-PP2B mice in the free exploration task (Figure 2C,E,G).

We agree with the reviewer that spatial map shift should similarly induce a re-exploration of the cue in control and mutant mice. Yet, in the control mice this phenotype was sporadic and only observed in three sessions from one single mouse, therefore the statistical power was too low for a strong interpretation of the corresponding behavioral data.

We have now improved the figure 2 by presenting scatter plots of all datapoints which allows for a more direct reading of the datasets and S1 and S2 separately (following minor point # 5), in addition to the ratios. Statistical analysis using non parametric Wilcoxon Signed-Ranks test and adjusted p values for multiple comparisons revealed an increase in object exploratory behavior in S2 relative to S1 during sessions with spatial map rotation in L7-PP2B mice, whereas this was not the case during stable sessions (mean distance from the cue: stable sessions $Z=0.82$, $p=0.82$; spatial map rotation, $Z=3.24$, $p=0.0024$; % of time in the cue zone: stable sessions $Z=0.20$, $p=1$; spatial map rotation, $Z=2.37$, $p=0.0368$; normalized number of entries in the cue zone: stable sessions $Z=0.017$, $p=1$; spatial map rotation, $Z=2.77$, $p=0.011$). Although no difference was found in object behavior during spatial map rotation in the control group (mean distance from the cue: stable sessions $Z=1.59$, $p=0.22$; spatial map rotation, $Z=1.07$, $p=0.58$; % of time in the cue zone: stable sessions $Z=1.47$, $p=0.284$; spatial map rotation, $Z=1.60$, $p=0.22$; normalized number of entries in the cue zone: stable sessions $Z=2.38$, $p=0.035$; spatial map rotation, $Z=1.60$, $p=0.221$), this lack of significance should be interpreted with caution and might simply result from the low sample size. Indeed, the graph shows a clear tendency of the control as well to explore more the object during unstable sessions compared to stable ones. This analysis therefore supports our conclusions that spatial map instability leads to disorientation.

3) Second, the impairments in watermaze performance of L7-PP2B mice are very subtle. In fact, L7-PP2B mice learn to find the hidden platform by swimming the same distance as controls (Figure 4A),

and the deficits reported only show up as small differences in the initial segment of their swim trajectory (Figure 4B-E). It is peculiar that these subtle deficits are present from the first session of training (S1) presumably before learning has taken place. These effects may be more consistent with a subtle deficit in motor performance than a deficit caused by spatial disorientation. Given that L7-PP2B mice have impairments in basic motor function of vestibular reflexes (Schonewille et al., Neuron 2010), it is critical that the authors determine the root cause of the subtle behavioral impairments reported in the watermazetask.

This is an important point as it suggests that our data have been misinterpreted, presumably due to a lack of precisions regarding the behavioral analyses presented in this figure.

For instance, the reviewer states that *“the deficits reported only show up as small differences in the initial segment of their swim trajectory (Figure 4B-E).”* In fact, the analyses performed on the initial segment are represented only in Figure 4C. The panels B, D and E represent analyses performed on the entire trajectories. The analyses presented in Figure 4C that focused on mice location soon after they started to swim demonstrated that L7-PP2B mice initially swim more often than controls in the wrong direction. In contrast, Figure 4E presents the location of the focused research areas (the peak of single-trial exploration map computed on their entire trajectories) and shows that these are more distant to the platform, i.e. less relevant. These results are coherent with our findings showing an altered stability of the spatial maporientation.

In addition, in L7-PP2B mice the higher proportion of long paths observed in the Morris Water Maze (Fig. 4B) suggests the presence of failed trials even though the mice have learned the task. In addition to these results we have now also inspected variability of mice performances by analyzing the number of transitions between successful and failed trials once the task is learned. We found that L7-PP2B mice performances were indeed more variable as illustrated by the higher successful/failed trials transitions during the last 2 days of training (i.e. when the performances of control mice reached a plateau) (see revised figure 4 panel C). This new result is now added in the revised manuscript.

Overall, we acknowledge that the different behavioral parameters used to measure navigation performances in figure 4 lacked some explanations and we have now improved their description in the appropriate method (p.12-13), the legend of the figure (Fig. 5) and we also added an illustration of the parameters (insets) in the figures as well as a supplementary figure describing these analyses (Fig. S10).

The reviewer also gathers that *“It is peculiar that these subtle deficits are present from the first session of training (S1) presumably before learning has taken place. These effects may be more consistent with a subtle deficit in motor performance than a deficit caused by spatial disorientation.”* Importantly, the first session of training shows no statistical difference between control and L7-PP2B mice in all behavioral analyses performed (Fig. 4 A-E). In addition, to ensure that the deficits observed in subsequent sessions were not related to motor impairments we performed correlational analysis between L7-PP2B swimming speed and watermaze parameters that revealed the mutant phenotype such as “mouse-platform distance after initial segments of trajectory” and the “distance peak of exploration- platform”. Our results indicate that none of these parameters correlated with speed (see figure below) thus suggesting that the deficits observed in the L7-PP2B mice in the Morris Water Maze do not result from impaired motor performances. These new results are now added in the revised manuscript (Fig. S9C and D). Please also note that our animals were younger than 6 months of age, and that motor performance deficits in L7-PP2B mice usually only start to dominate at ages beyond 9 months, at which their Purkinje cells may show signs of degeneration (De Zeeuw, personal communication).

No correlation was found between L7-PP2B spatial performances and speed.

4) For example, since the authors claim that impaired performance in the watermaze is the result of unstable hippocampal maps that rotate between different exposures and cause disorientation, it would be useful to compare the watermaze performance of L7-PP2B mice with two other control groups: (1) normal wildtype mice that are disoriented before each training session and (presumably) have unstable place field representations, (2) normal wildtype mice that are trained in an unstable environment that “rotates” from session to session (or alternatively, an unstable environment in which the hidden platform moves from session to session according to a fixed random rotation).

The reviewer suggests two additional Water Maze experiments to investigate the link between spatial map disorientation and altered navigation performances. Although our study was not designed to directly evaluate the impact of place cells instability on navigation behavior in the Morris Water Maze, we performed the first experiment proposed by the reviewer, with the mindset of providing experimental elements that might foster deeper discussion on this point. Thus, we assessed the behavioral performances of wild-type mice that have been disoriented before each training session in the Water Maze. Our results, originating from two independent sets of experiments, indicated that disoriented mice do not show deficit in the Water Maze and even tend to learn faster to find the platform than non-disoriented controls. This is potentially due to an over-reliance on external cues when vestibular information is disturbed (see figure below).

These new results are in sharp contrast with the phenotype of L7-PP2B mice (Fig. 5 of the manuscript), highlighting that the deficit observed in the latter is not equivalent to an externally applied disorientation leading to vestibular perturbations. In fact, these results suggest that impaired spatial map and navigation performances of L7-PP2B mice may emerge from the alteration of multimodal sensory processing by the cerebellum rather than from vestibular perturbations.

Effect of a manual disorientation on control mice performances in the Morris Water Maze

Finally, we do not understand the second experiment proposed by the reviewer. How can mice find a hidden platform using external cues if the position of the platform (or the cues) moves from session to session? What results would be expected in such conditions?

5) Lastly, it is critical that the authors evaluate whether the place fields of L7-PP2B mice are unstable under conditions that more closely resemble what the mice experience in the watermaze task (which include multiple prominent distal cues and are very different from the conditions experienced during free exploration in a stark arena with a single proximal cue). Without this kind of information it is impossible to make any claims about what may be causing the subtle deficits of L7-PP2B mice in the watermaze task, or whether their place cell representations are unstable in this task at all.

The aim of our study is to determine how opposite forms of plasticity at cerebellar Purkinje cells differentially affect the spatial code. Thus, the rationale for using an arena with a single proximal cue was to analyze L7-PP2B mice in conditions similar to the ones that revealed the L7-PKCI phenotype. We besides performed experiments in the Morris Water Maze in order to test whether, in accordance with the deficits in spatial representation we unravel, navigation of L7-PP2B mice might be impaired.

We have now added new analyses and figures (Fig. S10) that better describe the impairments of L7-PP2B mice in the Morris Water Maze. New figure 5 illustrates that they exhibit a peculiar phenotype with more variable performances, altered research behavior and inaccurate initial orientation. As the reviewer rightly points out, the deficits observed in the Water Maze task may not be directly linked to their place cells instability. Yet, insofar as the spatial navigation impairments of L7PP2B mice are compatible with their altered hippocampal representation, we believe they remain highly relevant to our study. Thus, we have carefully rewritten the manuscript to make it clear that we describe how a PP2B deficiency in the cerebellum impacts the hippocampal representation on the one hand and spatial navigation on the other.

MINOR POINTS:

1) Specify whether experiments were run blind as to genotype. If not, what kind of precautions were taken to make sure that all mice were handled in exactly the same way? This is particularly critical for the place cell experiments, given that the phenotype reported is a place field rotation that can occur in both wildtype and L7-PP2B mice.

Yes, both electrophysiological and behavioral experiments were run blind as to the genotype. We thank the reviewer for pointing out this omission. We have now included this point in the appropriate method section (p.1, lines 7-8)

2) Specify how many S1-S2 sessions were experienced by each mouse. If more than one, was the location of the cue changed between one S1-S2 session and the next S1-S2 session?

Several sessions were experienced by each mouse (ranging from 2 to 16) and the location of the cue was always at the same position between different S1-S2 sessions. We have now added this point in appropriate method section (p. 3, lines 20-21)

3) Use same format as in Figure 3I (right panel) to specify number of cells for each mouse in Figure 1D,E.

We have now modified the figure 1D –E to use the same format than the one used in the Figure 3I.

4) Figure 2C indicates that place cells were “Always stable” in 4 out of 7 of the L7-PP2B mice. A statistical analysis is necessary to determine whether this is a consequence of the probabilistic (sporadic) nature of the place field rotations, or whether the phenotype is absent in a population of L7-PP2B mice. If the place cell instability phenotype is really absent in many of the L7-PP2B mice, it will be critical to evaluate performance in the watermaze separately for L7-PP2B mice with and without the phenotype.

We thank the referee for raising this point. We wish to highlight that the previous analysis underestimated the L7-PP2B phenotype. Analysis of instability has now been improved in two ways. First we have improved the dataset by adding electrophysiological data from both control and mutant mice. Second, we have improved the criteria used to classify a mouse as stable or unstable since the

classification previously performed artificially underestimated the phenotype. Indeed, considering that we have an occurrence of instability of ~25 % (see revised Fig. 1), a mouse may be classified as stable or unstable only if more than 4 recording sessions have been collected. Out of the 6 L7-PP2B mice reaching this criterion, 5 show instability.

These proportions are thus more likely the consequence of the probabilistic nature of place field rotations rather than a phenotype that would be present only in a subgroup of L7-PP2B mice. We have consistently modified the presentation of behavioral data in Figure 2.

5) Specify the results in Figure 2 separately for S1 and S2, not as a ratio S1/S2. In addition to the bars showing the average value and the standard deviation in each one of the plots, show the individual points that comprise the average.

According to reviewer's request, we have modified the figure to present individual data and specified separately the data for S1 and S2 in addition to the ratio (see revised Fig. 2).

Reviewers' comments:

Reviewer #1 (Remarks to the Author):

The authors have answered many of my concerns. I do however, have a few remarks left, most of which are minor. My only main concern remaining is that I am not sure how robust the data is based on the sample size. For example, in figure S6, panel B, does each session also represent unique cells? If so, what methods were used to ensure that these cells were indeed different? In essence, I'd like to know that the instability was observed across multiple knockout animals (which the authors show is the case) and across multiple *unique* cells (rather than sessions) within these knockout animals (which is still a little unclear to me). A table in the supplement for example, could clear this up.

Other minor comments:

I'm a bit confused about the use of 'intra-session stability'. Page 5, line 22; I assume you mean stability within one session but on Page 6, line 2, you use the same term to describe stability between S1 and S2. Wouldn't that be across-session stability? Or if not, why does the inter-session stability change between these two analyses?

Page 3, Line 10: rotation *has*

Page 4, Line 2: head-direction *cell* system

Reviewer #2 (Remarks to the Author):

The authors have satisfactorily addressed my previous concerns. However, I do have a few remaining comments about the revised manuscript. In addition, the manuscript still contains quite some grammatical errors and will require copyediting.

1. On page 5, line 21, the authors should change "marginal" to "non-significant".
2. On page 6, line 1, the authors should avoid exaggerated language. Remove "drastic".
3. On line 18, page 8, the meaning of the following passage is unclear: "among which 13 sessions originated from ensemble recordings". How was spontaneous representation assessed in ensembles? Wasn't this a measure that was calculated for individual units?
4. On page 20, line 9, inappropriate references are provided. Vinueza-Veloz has nothing to do with gamma. Bragin et al. did not calculate coherence between hippocampus and entorhinal cortex; no entorhinal cortex recordings were performed in that study. The proper reference is Colgin et al. 2009.

Reviewer #3 (Remarks to the Author):

The revised manuscript falls short of addressing my original concerns. I have listed those original concerns below, and explained why the authors' revisions did not address them satisfactorily:

- 1) I pointed out in my original review that it was CRITICAL to rule out the possibility that navigation impairments in L7-PP2B are caused by problems in self-motion spatial representations. In the rebuttal, the authors agree with my point. I suggested that the authors perform new experiments to assess watermaze performance in the dark. Normal performance in finding the hidden platform in the dark would support the authors' claim that L7-PP2B mice have normal self-motion representations and that they can use these representations to navigate in the dark.

The revised manuscript does not show any data for watermaze performance in the dark. In their rebuttal, the authors mention that they tested the abilities of L7-PP2B mice to navigate toward a goal using self-motion information. They provide a figure in the rebuttal and point out that they found no difference between control and L7-PP2B navigation performance in a self-motion based version of the Morris Water Maze. However, there are no details about how this experiment was performed, or what analyses were done. It is stupefying that this critical experiment is relegated to the rebuttal, and that it is not even mentioned in the manuscript.

As I already indicated in my original review, it is critical to perform the watermaze experiment in the dark in exactly the same way as the authors performed the same experiment in their previous work (Science, 2011). It is also critical that the data for assessing performance in the dark is analyzed in exactly the same way as the data for assessing performance in the light (Figure 5), including analyses of bimodality in traveled distances (Figure 5B), success/fail transitions (Figure 5C), initial orientation (Figure 5D), and mean peak distance (Figure 5E). These direct comparisons are required for making any claims about the ability of L7-PP2B to rely on self-motion representations for navigation. In this context, it is quite disconcerting that in terms of Total travelled distance, the general performance of both control and L7-PP2B appears to be very different for the data in Figure 5 (Figure 5A; travelled distance goes from 1200cm at the beginning of training to 500cm at the end of training) and for the data in the figure shown in the rebuttal (Rebuttal figure; 600cm at the beginning of training to 200cm at the end of training).

2) I pointed out in my original review that some of the results appeared to be inconsistent with the authors' major claim (e.g., see Title of manuscript) that impaired navigation performance in L7-PP2B mice is the result of spatial disorientation. To address this point, I suggested that the authors perform new experiments to compare watermaze performance in L7-PP2B mice and in control mice that are disoriented prior to testing their navigation ability.

The authors have performed the experiment I suggested, but the results are exactly the opposite of what would be expected if their interpretation were correct. The authors report in the rebuttal that when control mice are disoriented prior to the watermaze test, they do not show deficits and even tend to learn to find the platform faster than non-disoriented controls. As the authors admit in their rebuttal, the results obtained in the disoriented control mice are in sharp contrast with the phenotype of L7-PP2B mice (Figure 5), clearly demonstrating that the deficit observed in the latter is not equivalent to an externally applied disorientation. This is an important finding and should be included in the manuscript. I do not understand why it is only mentioned in the rebuttal.

Given the new data, which demonstrates that the navigation impairments of L7-PP2B mice cannot be attributed to disorientation, there are instances throughout the paper where the authors' claims appear unwarranted. For example, the title states "Impaired cerebellar Purkinje cell potentiation generates spatial disorientation". Similar claims can be found in the One sentence-summary, and in the Discussion. There is nothing in the paper to support these claims about the role of disorientation in L7-PP2B mice. Instead, the new data clearly demonstrates that impaired cerebellar Purkinje cell potentiation results in navigation problems via deficits in neural mechanisms that are quite separate and distinct from those which occur during disorientation. What those distinct mechanistic impairments may be is not addressed, and this remains a major shortcoming of the current manuscript.

To try to gain mechanistic insight, and address what may be causing the deficits of L7-PP2B mice in the watermaze task, I had made two recommendations in my previous review:

First, I had suggested the possibility that the impairments in watermaze navigation may reflect a subtle deficit in motor performance rather than a deficit caused by spatial disorientation. I pointed out that L7-PP2B mice have impairments in basic motor function of vestibular reflexes, as reported by Schonewille et al., Neuron 2010. In their rebuttal, the authors note that this kind of impaired

vestibular function is unlikely to play a role in the observed impairments because their animals were younger than 6 months of age, and motor performance deficits in L7-PP2B mice usually only start at ages beyond 9 months. This is incorrect. The mice with impaired vestibular function in the Schonewille et al. article were 10-26 weeks. The manuscript would benefit greatly from an in-depth analysis of vestibular function in the mice, including analysis of head-direction cells (see Reviewer #1 for similar comments), which have been shown to play a role in generating unstable place cell representations during disorientation (Knierim et al., J Neurosci 1995).

Second, I had indicated that it was critical for the authors to evaluate whether the place fields of L7-PP2B mice are unstable under conditions that more closely resemble what the mice experience in the watermaze task (which include multiple prominent distal cues and are very different from the conditions experienced during free exploration in a stark arena with a single proximal cue). The authors did not perform any new experiments to address this point, and therefore it is impossible to make any claims about the underlying cause for the navigation impairments observed in the watermaze. We do not even know if place cell representations are unstable in the watermaze task at all. The lack of any mechanistic insight is a major shortcoming of the current manuscript.

MINOR POINTS:

- 1) When reporting the number of stable and unstable recording sessions, please indicate the fraction of unstable sessions per mouse. For example, one of the control mice had 3 unstable recording sessions; 3 out of how many total recording sessions for this mouse?
- 2) Issues with Figure S6:
 - i) The labels in Figure S6 indicate that some of the data came from session 136 (M1247s136) but the total number of recording sessions reported on page 8 was 130 ($57 + 73 = 130$).
 - ii) In Figure S6, there are 14 unstable sessions with multiple cells recorded but the text on page 8 indicates that there were only 13 unstable sessions with multiple cells recorded.
 - iii) Finally, from the labels of the sessions in Figure S6, it seems that there were particular sessions in which instability was likely to occur. For example, session 56 had unstable maps for M723 and M696. Session 31 had unstable maps for M723 and M801. Session 16 had unstable maps for M801 and M802. Session 36 had unstable maps for M801 and M802. Is this just a coincidence or was something different about those particular sessions for which more than one mouse was found to have unstable place field representations?
 - iv) The average angle rotation on unstable sessions looks (by eye) to be quite consistent within the same mouse. For example, the angle rotation for the 3 unstable sessions recorded in M802 seems to be very close to 150 for all 3 sessions. The text on page 8 claims that several different rotation angles of mouse spatial maps were found both within and between mice. Can this claim be quantified in more detail? For example, is the distribution of rotation angles within and between mice differ from a uniform random distribution?

Reviewers' comments:

Reviewer #1 (Remarks to the Author):

We thank the reviewer for his/her positive statement. We have now addressed all the remaining points raised by the reviewer (see below).

The authors have answered many of my concerns. I do however, have a few remarks left, most of which are minor. My only main concern remaining is that I am not sure how robust the data is based on the sample size. For example, in figure S6, panel B, does each session also represent unique cells? If so, what methods were used to ensure that these cells were indeed different? In essence, I'd like to know that the instability was observed across multiple knockout animals (which the authors show is the case) and across multiple *unique* cells (rather than sessions) within these knockout animals (which is still a little unclear to me). A table in the supplement for example, could clear this up.

The instability was observed across 5 different L7-PP2B mice (Fig. 1E, S6 and S7) and across 47 cells distributed in 18 different recording sessions. We ensured that each unstable session contains cells that are different for the previous and subsequent unstable sessions by lowering tetrodes between different recording days (now mentioned in the supplementary section page 4 lines 2-3) and comparing the waveforms profiles on each electrode of the tetrodes. This allows us to say that 46 out of the 47 unstable cells are most certainly unique cells. On one exceptional case however, a same cell has been recorded on 2 consecutive days displaying instability (801s31t0a and 801s36t0a). Importantly instability was not a characteristic of a subset of cells, but corresponded to events that, when they occurred, affected the whole population of hippocampal place cells. To clarify this point we have now indicated the number of different unstable cells per mouse in a supplementary table (new Fig. S6).

Other minor comments:

I'm a bit confused about the use of 'intra-session stability'. Page 5, line 22; I assume you mean stability within one session but on Page 6, line 2, you use the same term to describe stability between S1 and S2. Wouldn't that be across-session stability? Or if not, why does the inter-session stability change between these two analyses?

Indeed, intra-session stability mentioned on page 5 line 22 described stability within one given session. This parameter is not affected in L7-PP2B mice. However, inter-session stability described on page 6 line 1-2 refers to stability across sessions, i.e. between session 1 (S1) and session 2 (S2). This parameter is altered in L7-PP2B mice. To avoid any confusion, we have now added "stability within a session" page 5 line 22.

Page 3, Line 10: rotation *has*

We have now corrected this grammatical error (page 3 line 10).

Page 4, Line 2: head-direction *cell* system

We have now corrected this grammatical error (page 4 line 2).

Reviewer #2 (Remarks to the Author):

We thank the reviewer for the positive statement. We have now addressed all the remaining points raised by the reviewer (see below).

The authors have satisfactorily addressed my previous concerns. However, I do have a few remaining comments about the revised manuscript. In addition, the manuscript still contains quite some grammatical errors and will require copyediting.

1. On page 5, line 21, the authors should change “marginal” to “non-significant”.

We have now replaced “marginal” with “non-significant” (page 5 line 20-21).

2. On page 6, line 1, the authors should avoid exaggerated language. Remove “drastic”.

We have now removed “drastic” (page 6 line 1)

3. On line 18, page 8, the meaning of the following passage is unclear: “among which 13 sessions originated from ensemble recordings”. How was spontaneous representation assessed in ensembles? Wasn’t this a measure that was calculated for individual units?

Indeed, instability has been evaluated for each individual unit, and is characterized by a low S1-S2 coefficient and a large S1-S2 rotation angle. This instability has been observed in sessions during which one place cell was recorded and in sessions during which multiple place cells were recorded (and were all unstable together). The passage mentioned by the reviewer only indicates the proportion of recordings during which several unstable place cells were recorded simultaneously. To make this point clearer, we have now replaced this passage by “among which 14 sessions included multiple place cells” (page 8, line 18).

4. On page 20, line 9, inappropriate references are provided. Vinuesa-Veloz has nothing to do with gamma. Bragin et al. did not calculate coherence between hippocampus and entorhinal cortex; no entorhinal cortex recordings were performed in that study. The proper reference is Colgin et al. 2009.

We thank the reviewer for pointing out this inaccuracy. The reference Bragin et al has now been removed from the sentence (page 20, line 19) and reference numbers are now associated with the correct reference.

Reviewer #3 (Remarks to the Author):

The revised manuscript falls short of addressing my original concerns. I have listed those original concerns below, and explained why the authors' revisions did not address them satisfactorily:

1) a) I pointed out in my original review that it was CRITICAL to rule out the possibility that navigation impairments in L7-PP2B are caused by problems in self-motion spatial representations. In the rebuttal, the authors agree with my point. I suggested that the authors perform new experiments to assess watermaze performance in the dark. Normal performance in finding the hidden platform in the dark would support the authors' claim that L7-PP2B mice have normal self-motion representations and that they can use these representations to navigate in the dark.

The revised manuscript does not show any data for watermaze performance in the dark. In their rebuttal, the authors mention that they tested the abilities of L7-PP2B mice to navigate toward a goal using self-motion information. They provide a figure in the rebuttal and point out that they found no difference between control and L7-PP2B navigation performance in a self-motion based version of the Morris Water Maze. However, there are no details about how this experiment was performed, or what analyses were done. It is stupefying that this critical experiment is relegated to the rebuttal, and that it is not even mentioned in the manuscript.

As I already indicated in my original review, it is critical to perform the watermaze experiment in the dark in exactly the same way as the authors performed the same experiment in their previous work (Science, 2011). It is also critical that the data for assessing performance in the dark is analyzed in exactly the same way as the data for assessing performance in the light (Figure 5), including analyses of bimodality in traveled distances (Figure 5B), success/fail transitions (Figure 5C), initial orientation (Figure 5D), and mean peak distance (Figure 5E). These direct comparisons are required for making any claims about the ability of L7-PP2B to rely on self-motion representations for navigation.

The aim of our study was to evaluate how hippocampal dependent representation and goal-directed navigation are affected following a deficit in cerebellar PP2B dependent potentiation. Therefore, we first analyzed hippocampal place cell properties in two different sensory conditions, light and dark, to assess allocentric versus self-motion hippocampal representation. According to the deficit we found, in this case an incorrect anchoring of the spatial map on the allocentric cue, we then tested allocentric navigation performances of L7-PP2B mice. The absence of place cell deficit in the dark indicated an intact self-motion dependent hippocampal representation (Fig. 3). This is the reason why we did not deeply analyze self-motion goal-directed navigation in the first revision of our manuscript.

However, considering reviewer 3 viewpoint, we understand his/her request to 1) have a complete picture of L7-PP2B hippocampal representation and navigation in both allocentric and self-motion conditions and 2) compare L7-PP2B performances with the ones previously obtained in L7-PKCI (Rochefort et al., 2011).. We have thus carefully examined the ability of L7PP2B mice to navigate toward a goal in the dark i.e. with low external information and highly relevant self-motion information (Fig. S12).

In the preliminary results we sent in our previous rebuttal, we used the exact same protocol than the one used in the Science. However, the minimum distance reached by the control mice of the L7-PP2B project was higher compared to L7-PKCI controls (250 vs 100 cm respectively, compare the figure from previous rebuttal with the left panel of the Figure 1 below). In the new set of experiments, we have extended the number of training sessions to take this difference into account and obtain comparable performances between the two types of controls (compare left and middle panel below). The first tentative to assess L7-PP2B performances in the dark unfortunately revealed an inability for control mice to reach the same level as L7-PKCI mice (dark sessions of the 1st training, see middle panel below, ANOVA, session effect, $F_{1,5}=1.28$, $p=0.31$). We therefore performed additional trials to get dark performances of L7-PP2B controls similar to the L7-PKCI controls (2nd training, right panel below, ANOVA, session effect, $F_{1,5}=12.61$, $p<0.001$).

Figure 1. Comparisons of L7-PKCI and L7-PP2B performances in the dark version of the Morris Water Maze.

The 2nd training was then analyzed “in exactly the same way as the data for assessing performance in the light (spatial Water maze, Figure 5)” as requested by the reviewer (see figure 2 below). Our results indicate that L7-PP2B mice were impaired in the dark compared to controls as illustrated by the increased in total distance ($F_{1,18}=5.06$, $p=0.037$) and altered initial orientation ($F_{1,18}=5.61$, $p=0.029$, group effect, ANOVA with repeated measures). The mean peak distance and the success/fail transitions were similar between control and L7-PP2B mice (mean peak distance, $F_{1,18}=2.75$, $p=0.12$, ANOVA with repeated measure; success/fail transition : $p=0.47$, Mann-Whitney U-test, see Figure 2 below). It is important to mention that the mean peak distance, which measures the distance between the peak of exploration during the trajectory and the position of the platform has been developed to analyze complex trajectories varying from trials to trials (with different departure points between each trials). This parameter is not suitable for analyzing short and stereotyped trajectories and is thus not presented in the revised manuscript.

Figure 2. Performances of L7-PP2B mice in the self-motion based navigation task assessed in exactly the same way as the data for assessing performance in the light (spatial Water maze, Figure 5).

We further complemented our analysis by assessing mice heading as well as abilities to perform direct trajectories. Our data indicate that these two parameters are also affected in L7-PPB mice in the dark specifically (see below and Fig. S12 from the revised manuscript).

Together, our data suggest that L7-PP2B mice are impaired in using self-motion information to optimize their path toward a goal. Importantly, the fact that hippocampal spatial representation of L7-PP2B mice is intact in the dark (Fig 3) suggest that the deficit observed here is not the consequence of an altered self-motion based spatial representation in the hippocampus. This finding is particularly interesting as it allows to dissociate the influence of cerebellar plasticity on hippocampal activity on the one hand, and on goal-directed behavior on the other hand. Interestingly, this idea fits with our previous findings in L7-PCI mice which displayed intact allocentric hippocampal representation but altered allocentric navigation performances (see Rochefort et al., 2011 and Burguiere et al., 2005 respectively).

In light of our previous findings in L7-PCKI mice, we have now converging evidences suggesting that the cerebellum participates to hippocampal spatial representation through different mechanisms with PKC dependent processes involved in self-motion based hippocampal spatial representation (Rochefort et al., 2011) and PP2B dependent ones crucial for allocentric anchoring of hippocampal representation (present data) . Second, spatial navigation behavior is affected in both visual and self-motion conditions showing that the cerebellum has a part of direct influence on navigation behavior independently of the type of plasticity involved (present data; Burguiere et al 2005, Rochefort et al., 2011)). We have now rewritten the manuscript to include and discuss this new finding (se for instance pages 18-19 and p.22 line 1-8).

b) In this context, it is quite disconcerting that in terms of Total travelled distance, the general performance of both control and L7-PP2B appears to be very different for the data in Figure 5 (Figure 5A; travelled distance goes from 1200cm at the beginning of training to 500cm at the end of training) and for the data in the figure shown in the rebuttal (Rebuttal figure; 600cm at the beginning of training to 200cm at the end of training).

Mice travelled distances are very different between the two tasks simply because the tasks are highly different. The spatial version of the Morris Water Maze (Fig. 5) assesses the capacity of an animal to learn precise environmental locations based on the relationship between a goal and the surrounding cues. Mice are trained to find the platform from different departure points. The figure shown in the rebuttal and in the present manuscript (Fig. S12) assessed the ability of the mice to navigate using mainly self-motions cues. In this task, and as described in our previous paper (Rocheffort et al., 2011) mice are first trained to find a platform in the light. This platform is visible and at a fixed position relatively to the departure point. In addition the departure point contains parallel arms to guide the initial orientation. So, in the light, the trajectory is always the same and becomes rapidly short and stereotyped. The global travelled distance is thus shorter than in the classical spatial version of the Morris Water Maze.

2) I pointed out in my original review that some of the results appeared to be inconsistent with the authors' major claim (e.g., see Title of manuscript) that impaired navigation performance in L7-PP2B mice is the result of spatial disorientation. To address this point, I suggested that the authors perform new experiments to compare watermaze performance in L7-PP2B mice and in control mice that are disoriented prior to testing their navigation ability. The authors have performed the experiment I suggested, but the results are exactly the opposite of what would be expected if their interpretation were correct. The authors report in the rebuttal that when control mice are disoriented prior to the watermaze test, they do not show deficits and even tend to learn to find the platform faster than non-disoriented controls. As the authors admit in their rebuttal, the results obtained in the disoriented control mice are in sharp contrast with the phenotype of L7-PP2B mice (Figure 5), clearly demonstrating that the deficit observed in the latter is not equivalent to an externally applied disorientation. This is an important finding and should be included in the manuscript. I do not understand why it is only mentioned in the rebuttal. Given the new data, which demonstrates that the navigation impairments of L7-PP2B mice cannot be attributed to disorientation, there are instances throughout the paper where the authors' claims appear unwarranted. For example, the title states "Impaired cerebellar Purkinje cell potentiation generates spatial disorientation". Similar claims can be found in the One sentence-summary, and in the Discussion. There is nothing in the paper to support these claims about the role of disorientation in L7-PP2B mice. Instead, the new data clearly demonstrates that impaired cerebellar Purkinje cell potentiation results in navigation problems via deficits in neural mechanisms that are quite separate and distinct from those which occur during disorientation. What those distinct mechanistic impairments may be is not addressed, and this remains a major shortcoming of the current manuscript. To try to gain mechanistic insight, and address what may be causing the deficits of L7-PP2B mice in the watermaze task, I had made two recommendations in my previous review:

First, I had suggested the possibility that the impairments in watermaze navigation may reflect a subtle deficit in motor performance rather than a deficit caused by spatial disorientation. I pointed out that L7-PP2B mice have impairments in basic motor function of vestibular reflexes, as reported

by Schonewille et al., Neuron 2010. In their rebuttal, the authors note that this kind of impaired vestibular function is unlikely to play a role in the observed impairments because their animals were younger than 6 months of age, and motor performance deficits in L7-PP2B mice usually only start at ages beyond 9 months. This is incorrect. The mice with impaired vestibular function in the Schonewille et al. article were 10-26 weeks. The manuscript would benefit greatly from an in-depth analysis of vestibular function in the mice, including analysis of head-direction cells (see Reviewer #1 for similar comments), which have been shown to play a role in generating unstable place cell representations during disorientation (Knierim et al., J Neurosci 1995).

Second, I had indicated that it was critical for the authors to evaluate whether the place fields of L7-PP2B mice are unstable under conditions that more closely resemble what the mice experience in the watermaze task (which include multiple prominent distal cues and are very different from the conditions experienced during free exploration in a stark arena with a single proximal cue). The authors did not perform any new experiments to address this point, and therefore it is impossible to make any claims about the underlying cause for the navigation impairments observed in the watermaze. We do not even know if place cell representation is unstable in the watermaze task at all. The lack of any mechanistic insight is a major shortcoming of the current manuscript.

We acknowledge that the preserved performances of the disoriented control mice in the Morris Water Maze task may seem puzzling at first sight. Indeed, one could theoretically expect that mice that have been externally disoriented might have impaired performances in a goal-directed spatial navigation task. However, the data we obtained in disoriented control mice confirmed previous data from the literature showing that externally disoriented mice do not show any impairment in the Morris Water Maze (Dudchenko et al., 1997, Martin et al., 1997, Gibson et al., 2001). Importantly, our data illustrate that the deficit of L7-PP2B mice is not equivalent to an externally applied disorientation, but may rather emerge from the alteration of internal processing by the cerebellum affecting the integration of different sources of sensory information such as vestibular, proprioceptive or visuo-motor information (Dokka et al., 2015, Laurens et al., 2013, Yakusheva et al., 2007 and 2013, Angelaki et al., 2010, Brooks et al., 2015 ...). As requested, these data are now incorporated in the revised manuscript (page 18, lines 8-14, and Fig. S11).

Regarding the additional experience suggested by the referee, we have doubt about its feasibility to test our main hypothesis. Indeed, the goal of our study is to understand if and how cerebellar activity might influence hippocampal spatial coding. We found that a lack of Purkinje cell PP2B-dependent potentiation alters the ability of the hippocampal map to maintain a stable orientation in an allocentric reference frame. The deficit was evidenced here precisely because allocentric orientation was based on a single navigational beacon (Chan et al., 2012, Knierim et al., 1995). It is highly conceivable that adding multiple prominent distal cues will facilitate allocentric-based spatial map anchoring and therefore hide the deficit of spatial map stability observed in L7-PP2B mice. Indeed, previous studies have shown that Head direction cell activity could be experimentally biased when rats navigated in a familiar environment containing few visual cue but their representation was not altered when they navigated in a cue-rich environment (Coletta et al., 2018). In addition, our new finding showing that L7-PP2B mice have intact self-motion dependent hippocampal representation but impaired self-motion goal-directed navigation behavior, do not orient toward a causal link between hippocampal place cell instability and the deficits observed in the Water Maze, but they rather describe two aspects of a phenotype.

The additional point raised by the referee concerns the interpretation of our results in term of mechanistic. The referee hypothesis is that the observed deficits could come from subtle deficits in motor performances, in particular through *basic motor function of vestibular reflexes*.

To answer reviewer's concern, we previously performed analyses demonstrating that impaired performances in the spatial version of the Morris Water Maze are not correlated with mouse speed and therefore are not linked with a difference in swimming ability (previous Fig S9C and D now Fig S10 C and D). It is important to remind that these correlational analyses had been added to a battery of sensory-motor task (measuring anxiety, locomotor activity, motor coordination, and motor balance) showing that all these parameters are preserved in L7-PP2B mice (Fig.S1).

To test the referee hypothesis further, we now also have examined vestibular function of L7-PP2B mice. We performed vestibular-dependent sensory-motor tasks in the dark following the protocol used in our previous studies (Rocheffort et al., 2011). Data are now reported in the Fig. S9 of the revised manuscript. L7-PP2B mice show intact equilibrium in the static and dynamic balance test, normal activity (rearing behavior) in the open field and normal motor adaptation in the rotarod test (FigS9A). These results, associated with the intact exploratory behavior of L7-PP2B mice in the dark (Fig S9 B-G) and with the intact hippocampal representation of L7-PP2B mice observed in the dark (Fig. **3**) do not suggest that the deficit observed in L7-PP2B mice could emerge from a basic alteration of the vestibular function. In addition, comparing the nature of the deficits observed in place cells of L7-PP2B in the light, this does not fit with the phenotype observed after inactivation of the vestibular system which led to the disruption of location-specific firing in hippocampal place cells (Stackman et al. (2002) or Russel et al (2003)). Together, these data do not favor referee's hypothesis of an alteration in the vestibular-HD pathways as an origin of the observed physiological or behavioral deficits. Furthermore, as previously answered to reviewer 1, it is important to have in mind that impaired cerebellar Purkinje cells activity can impact hippocampal activity through two major pathways: the vestibular nuclei-HD system on the one hand (see for example Smith, 2005) and the deep cerebellar nuclei-parietal cortex pathway on the other hand (Giannetti and Molinari, 2002). Exploring the HD cell pathway and/or the parietal pathway in L7-PP2B mice represents in itself a whole new investigation that will be eventually pursued in future dedicated studies.

Alternatively, the deficit described in L7-PP2B might result from an impaired integration of multisensory information. To test this hypothesis, we investigated hippocampal local field potential (LFP) activity in control and L7-PP2B mice during two consecutive sessions of free exploration of the familiar environment (new Fig. 4). LFP activity in theta (6-12 Hz) and low-gamma (20-45 Hz) bands was not different between controls and L7-PP2B mice (Fig 4B-C, theta: genotype, $F(1,8)= 1.24$, $p=0.30$; session, $F(1,8)= 2.44$, $p=0.16$; session * genotype interaction $F(1,8)= 0.31$, $p=0.59$; low gamma: genotype, $F(1,8)= 1.96$, $p=0.20$; session, $F(1,8)= 3.32$, $p=0.1$; session * genotype interaction $F(1,8)= 3.70$, $p=0.09$, ANOVA). However an increase in the high gamma band was observed in L7-PP2B mice compared to control mice in session S2 specifically (Fig 4B-C, genotype, $F(1,8)= 5.3$, $p=0.05$; session, $F(1,8)= 6.6$, $p=0.03$; session * genotype interaction, $F(1,8)= 6.1$, $p=0.04$; L7-PP2B, S1-S2, $p= 0.007$, controls-L7-PP2B in S2, $p=0.007$, LSD post-hoc test). This result point towards an increase in the computation linked with high gamma in the CA1 region of L7-PP2B mice specifically when mice re-enters the familiar environment. Gamma rhythms have been proposed to modulate interactions between substructures of the hippocampal formation such that low- and high-frequency gamma oscillations would mediate the coherence between CA1 and CA3 or entorhinal cortex (EC), respectively (Colgin et al., 2009). Thus, the low gamma/high gamma ratio in CA1 may reflect its input dominance toward a memory-based/CA3 versus a contextual/EC computational mode (Cabral et al., 2014). Here, the increased high gamma computation in L7-PP2B points toward an unbalance in favor of EC inputs

towards CA1 that may induce a bias in the integration of external sensory inputs, potentially leading to unstable anchoring of the spatial map to the allocentric frame (Fig. 4 and page 14-15).

In conclusion, what we propose in our new manuscript is the following. We agree with the referee that the origin of our deficit is distinct from those which occur during *external* disorientation. To avoid any confusion in the interpretation we removed the term disorientation from our manuscript. We instead made clear that the deficits we observed rather reflect internal deficient mechanisms related to the PP2B-dependent potentiation deficits in the cerebellum. Then in the light of previous and new experiments and analyses, we discuss the potential mechanistic origin of place cell instability by mentioning both, the vestibular hypothesis and the unreliable integration of multisensory information theory (page 20 lines 5-24).

MINOR POINTS:

1) When reporting the number of stable and unstable recording sessions, please indicate the fraction of unstable sessions per mouse. For example, one of the control mice had 3 unstable recording sessions; 3 out of how many total recording sessions for this mouse?

As suggested by the reviewer we have now indicated the fraction (percentage) of unstable session per mouse in a supplementary table that also contain the number of unstable cells recorded per mouse (following reviewer 1 suggestion) (see new Fig. S6)

2) Issues with Figure S6:

i) The labels in Figure S6 indicate that some of the data came from session 136 (M1247s136) but the total number of recording sessions reported on page 8 was 130 (57 + 73 = 130).

The session number indicated in the labeling is not an indication of the number of recording sessions included in the place cells analysis. Indeed, there are for each mouse several sessions during which no place cell was recorded or with an exploration that did not reach the covering threshold of 65 % (as mentioned in the supplementary method, page 2 line 16-17 "Starting 5 days after surgery, the activity from each tetrode was screened daily while the mice explored the recording cylinder" and page 5 line 11-12 "Data from a particular session were only accepted for analysis if more than 65% of the bins were covered by the animal").

ii) In Figure S6, there are 14 unstable sessions with multiple cells recorded but the text on page 8 indicates that there were only 13 unstable sessions with multiple cells recorded.

We have now corrected this in the text (see page 8 line 18).

iii) Finally, from the labels of the sessions in Figure S6, it seems that there were particular sessions in which instability was likely to occur. For example, session 56 had unstable maps for M723 and M696. Session 31 had unstable maps for M723 and M801. Session 16 had unstable maps for M801 and M802. Session 36 had unstable maps for M801 and M802. Is this just a coincidence or was something different about those particular sessions for which more than one mouse was found to have unstable place field representations?

This was just a coincidence. Indeed, these sessions were not particular.

iv) The average angle rotation on unstable sessions looks (by eye) to be quite consistent within the same mouse. For example, the angle rotation for the 3 unstable sessions recorded in M802 seems to be very close to 150 for all 3 sessions. The text on page 8 claims that several different rotation angles of mouse spatial maps were found both within and between mice. Can this claim be quantified in more detail? For example, is the distribution of rotation angles within and between mice differ from a uniform random distribution?

We thank the reviewer for this interesting suggestion. Indeed, the distribution of the median rotation angles highlights that within a same mouse, the rotation angles are relatively similar between different events (Fig. S7 C and D). The comparison with shuffle data indicates that this within-mouse-consistency is significant for mutant mice (no interpretation was made for the control group, since instabilities occurred only three times and in only one mouse). In contrast, median rotation angle vary from mouse to mouse (Fig 7E and F). We have thus modified the text accordingly (page 8 lines 19-20).

Reviewers' comments:

Reviewer #1 (Remarks to the Author):

The authors have addressed all of my concerns. I recommend publication.

Reviewer #2 (Remarks to the Author):

The authors have satisfactorily addressed my concerns.

However, the authors should perhaps tone down some of their conclusions. The comparisons between LTP and LTD are perhaps overstated; for example, this is mentioned in the abstract, implying that depression and potentiation will be directly compared in the present study which is not the case. Also, the implication that self motion-based information remains intact in the hippocampus of L7-PP2B mice is also perhaps overstated, as mentioned by Referee 3. The authors may also want to consider toning down some exaggerated language (e.g., "strongly suggests" on line 10, page 8 and "far below the value" on line 14, page 8).

Reviewer #3 (Remarks to the Author):

The authors have revised many of their original claims, based on new experiments and analyses that were requested in my previous reviews (see below). The new claims, which are in some cases completely opposite to the original claims, have significantly lessened the impact of the paper. In addition, there remain concerns about the interpretation of the results, particularly with regards to the source of the spatial navigation impairments in the L7-PP2B mice.

1) New experiments have demonstrated that the impairments of L7-PP2B mice are NOT caused by disorientation mechanisms.

A major claim in the original manuscript, as reflected in the original title, was that the impaired navigational function of L7-PP2B mice was caused by disorientation. At my request, the authors performed new experiments to evaluate this claim more rigorously. As the authors admit in their most recent rebuttal, "the new data illustrate that the deficit of L7-PP2B mice is NOT equivalent to an externally applied disorientation". The authors go on to speculate what, if not disorientation, could be the reason for the observed deficits in navigation. They suggest that the observed deficits in navigation could be caused by problems in multisensory integration, but no data is presented to support this hypothesis. As I already mentioned in my previous reviews, the lack of mechanistic insight as to the cause of the navigation impairments in L7-PP2B mice is a major shortcoming of the paper.

2) New experiments have demonstrated that L7-PP2B mice are impaired in navigation that relies on self-motion information.

A major claim in the original manuscript was that L7-PP2B mice had deficits in allocentric-based navigation, but not in self motion-based navigation. This dichotomy was the centerpiece of the paper, which still at this point continues to emphasize the "diverging" roles of cerebellar plasticity in L7-PKC mutants lacking Purkinje cell LTD (the authors' previous work), and in L7-PP2B mutants lacking Purkinje cell LTP (the authors's current work). At my request, the authors performed more experiments to evaluate this claim more rigorously. In their first rebuttal, the authors reported that L7-PP2B showed normal function in a self motion-based version of the Morris watermaze, but provided no details about how these experiments were performed or analyzed. Now, in the second rebuttal, they report the opposite result after performing additional experiments. The newly found impairments in self motion-based navigation demonstrate that the contribution of cerebellar LTP to navigation cannot be dissociated from the already published contribution of cerebellar LTD, profoundly reducing the impact of the manuscript (see "Significance" and Point #1 of my original review).

3) Vestibular problems in L7-PP2B mice

A major claim in the original manuscript was that the navigation deficits of L7-PP2B mice did not have a motor origin. As I pointed out in my first review, Schonewille et al. (Neuron 2010) have reported that L7-PP2B mice have impairments in basic motor function of vestibular reflexes. In the first rebuttal, the authors dismissed my concerns that these motor/vestibular impairments could contribute to navigation deficits in the L7-PP2B mice because they thought that the mice used in the Schonewille paper were much older than the mice they had used for their own experiments in the current paper. As far as I can tell, this is incorrect. As I indicated in my 2nd review, the mice used in the Schonewille paper were 10-26 weeks old, which is very similar to the age of the mice in the manuscript under consideration (2.5 months – 6 months). The authors have performed new experiments to assess vestibular function in their mice, and they now report that L7-PP2B mice have no deficits in vestibular function. Are the vestibular deficits reported in the Schonewille paper not present in their mice? Can the authors really rule out the simple explanation that navigation impairments in the L7-PP2B mice may have a vestibular origin? The impact of the paper would be much reduced if the navigation impairments of L7-PP2B mice could be ascribed to vestibular/motor problems, given the well-established role of the cerebellum in motor function.

Overall Evaluation:

The original paper suggested an interesting dichotomy between the roles of cerebellar LTP in allocentric-based navigation (examined in this paper using L7-PP2B mice), and cerebellar LTD in self motion-based navigation (examined using L7-PKC mice in a previous paper by the same authors). Although this dichotomy continues to be emphasized in the most recent version of the manuscript, new experiments and analyses have demonstrated that the contribution of cerebellar LTD and LTP to spatial navigation cannot be dissociated.

There does seem to be an important difference in the way that cerebellar LTD and LTP contribute to spatial representations in place cells of the hippocampus. In their previous paper, the authors used L7-PKC mice to show that Purkinje LTD is required for maintaining place cell stability that relies on self motion cues in the dark, but not for maintaining place cell stability that relies on allocentric visual cues in the light. In the current paper, the authors use the L7-PP2B mouse to show that Purkinje LTP is required for maintaining place cell stability in the light but not in the dark. This difference in place cell stability between L7-PKC and L7-PP2B mice is the most notable result of the paper, but its impact is reduced for the following reasons:

1) There is nothing in the paper that speaks to the behavioral consequences of the difference in place cell stability between the L7-PKC and the L7-PP2B mice. Indeed, the authors admit in their last rebuttal that there is not a causal link between place cell stability of the L7-PP2B mice and their performance during spatial navigation.

2) Previous work has reported that L7-PP2B mice have vestibular/motor problems (Schonewille et al., Neuron). Thus, the possibility that a simple problem in vestibular/motor function could be the origin of the observed impairments in spatial navigation in the L7-PP2B cannot be ruled out. Given the data presented, there is nothing to support the claim that impaired Purkinje LTP in the L7-PP2B mice leads to deficits in spatial navigation via effects on the hippocampus. If the spatial navigation deficits of the L7-PP2B were caused by vestibular/motor dysfunction (and not by hippocampus problems), the impact of such a finding would be reduced because the link between impaired cerebellar plasticity and motor dysfunction is well-known.

Reviewer #2 (Remarks to the Author):

The authors have satisfactorily addressed my concerns.

However, the authors should perhaps tone down some of their conclusions. The comparisons between LTP and LTD are perhaps overstated; for example, this is mentioned in the abstract, implying that depression and potentiation will be directly compared in the present study which is not the case. Also, the implication that self motion-based information remains intact in the hippocampus of L7-PP2B mice is also perhaps overstated, as mentioned by Referee 3. The authors may also want to consider toning down some exaggerated language (e.g., "strongly suggests" on line 10, page 8 and "far below the value" on line 14, page 8).

We thank the reviewer for his/her helpful comments throughout the different revisions of this manuscript. We have now modified the abstract. We also toned down some exaggerate language (see page 8 line 10, 14 and 21). We have also carefully weighted our vocabulary to avoid any confusion between self-motion information used during navigation on the one hand and for hippocampal representation on the other hand.

Reviewer #3 (Remarks to the Author):

The authors have revised many of their original claims, based on new experiments and analyses that were requested in my previous reviews (see below). The new claims, which are in some cases completely opposite to the original claims, have significantly lessened the impact of the paper. In addition, there remain concerns about the interpretation of the results, particularly with regards to the source of the spatial navigation impairments in the L7-PP2B mice.

1) New experiments have demonstrated that the impairments of L7-PP2B mice are NOT caused by disorientation mechanisms. A major claim in the original manuscript, as reflected in the original title, was that the impaired navigational function of L7-PP2B mice was caused by disorientation. At my request, the authors performed new experiments to evaluate this claim more rigorously. As the authors admit in their most recent rebuttal, "the new data illustrate that the deficit of L7-PP2B mice is NOT equivalent to an externally applied disorientation". The authors go on to speculate what, if not disorientation, could be the reason for the observed deficits in navigation. They suggest that the observed deficits in navigation could be caused by problems in multisensory integration, but no data is presented to support this hypothesis. As I already mentioned in my previous reviews, the lack of mechanistic insight as to the cause of the navigation impairments in L7-PP2B mice is a major shortcoming of the paper.

We deeply disagree with this point but, as mentioned in our previous letter, this could be due to a divergence with the semantic use of the terms "spatial disorientation". Our data indicate that L7-PP2B mice display an unstable orientation of the hippocampal spatial representation (Fig 1). In our manuscript, the term spatial disorientation describes this instability. The fact, that, in addition L7-PP2B mice spent more time exploring the familiar object specifically when their spatial map shifts, reinforces the idea, at the behavioral level of a spatial disorientation (Fig. 2). And, finally, the deficit of L7-PP2B mice in a goal-directed task illustrated by an impaired initial orientation and an increase in their performance variability reinforces this conclusion (Fig. 6). Our new experiments on externally disoriented C57/Bl6 mice performed following reviewer 3's request show that externally disoriented mice display a different behavior than the one observed in L7-PP2B mice (Supplementary Fig.11). These data clearly show that L7-PP2B mice deficit is not equivalent to an EXTERNAL disorientation (and thus to a vestibular perturbation), and thus point toward an internal cerebellar mechanism causing instable

hippocampal representation. We therefore do not understand the restrictive interpretation of “disorientation” made by the referee. Indeed, disorientation phenomena called topographical disorientation have been reported in some clinical cases, independently of external factors (Iaria and Burles, 2016) and the mechanism of such a topographical disorientation is still unknown.

Moreover as our original title was “Impaired cerebellar Purkinje cell potentiation generates spatial disorientation” the link raised by the reviewer between disorientation on the one hand and navigation on the other hand is improper. Rather, our title precisely indicated a link between a lack of PP2B in the cerebellum and spatial disorientation, which is observed at the level of the hippocampal spatial representation.

2) New experiments have demonstrated that L7-PP2B mice are impaired in navigation that relies on self-motion information. A major claim in the original manuscript was that L7-PP2B mice had deficits in allocentric-based navigation, but not in self motion-based navigation. This dichotomy was the centerpiece of the paper, which still at this point continues to emphasize the “diverging” roles of cerebellar plasticity in L7-PKC mutants lacking Purkinje cell LTD (the authors’ previous work), and in L7- PP2B mutants lacking Purkinje cell LTP (the authors’s current work). At my request, the authors performed more experiments to evaluate this claim more rigorously. In their first rebuttal, the authors reported that L7-PP2B showed normal function in a self motion-based version of the Morris watermaze, but provided no details about how these experiments were performed or analyzed. Now, in the second rebuttal, they report the opposite result after performing additional experiments. The newly found impairments in self motion-based navigation demonstrate that the contribution of cerebellar LTP to navigation cannot be dissociated from the already published contribution of cerebellar LTD, profoundly reducing the impact of the manuscript (see “Significance” and Point #1 of my original review).

This assertion is wrong and might result again from a terminology discrepancy. The reviewer uses the terms “navigation” and “representation” as if they were meaning the same thing. This is incorrect. Spatial navigation rely on a numerous other brain structures beyond the hippocampus (see for instance, Ekstrom et al., 2017; Dumont and Taube, 2015; Babayan et al., 2017). Therefore, mice can be affected in their ability to navigate optimally toward a goal without any alteration in their spatial hippocampal representation *per se*. We never claimed in our original manuscript that L7-PP2B had no deficit in self-motion based navigation. The data presented in our original manuscript focused on self-motion based hippocampal place cells firing properties and we always stated that L7-PP2B mice have no deficit in self-motion based hippocampal representation (Fig 3).

Moreover, the fact that, at the behavioral level, allocentric and self-motion based navigation are both altered without cerebellar LTD or LTP do not invalidate the key finding of the present manuscript showing that the contribution of cerebellar potentiation to hippocampal spatial representation is different from the contribution of cerebellar LTD (Rocheffort et al., 2011). In contrast, the new results and discussion clearly broaden the impact of our manuscript. Our results reveal that PP2B-dependent potentiation in cerebellar Purkinje cells contributes to maintain spatial representation of a familiar environment in a stable allocentric reference frame. Taking into consideration our previous findings in L7-PCKI mice, we have now converging evidences showing that the cerebellum participates to hippocampal spatial representation through different mechanisms with PKC dependent processes involved in self-motion based hippocampal spatial representation (Rocheffort et al., 2011) and PP2B dependent ones crucial for allocentric anchoring of hippocampal representation (present data) . Importantly, spatial navigation behavior is affected in both visual and self-motion conditions, thus showing that the cerebellum has a direct influence on navigation behavior independently of the type of plasticity involved (present data; Burguiere et al 2005, Rocheffort et al., 2011).

3) Vestibular problems in L7-PP2B mice. A major claim in the original manuscript was that the navigation deficits of L7-PP2B mice did not have a motor origin. As I pointed out in my first review, Schonewille et al. (Neuron 2010) have reported that L7-PP2B mice have impairments in basic motor function of vestibular reflexes. In the first rebuttal, the authors dismissed my concerns that these motor/vestibular impairments could contribute to navigation deficits in the L7-PP2B mice because they thought that the mice used in the Schonewille paper were much older than the mice they had used for their own experiments in the current paper. As far as I can tell, this is incorrect. As I indicated in my 2nd review, the mice used in the Schonewille paper were 10-26 weeks old, which is very similar to the age of the mice in the manuscript under consideration (2.5 months – 6 months). The authors have performed new experiments to assess vestibular function in their mice, and they now report that L7-PP2B mice have no deficits in vestibular function. Are the vestibular deficits reported in the Schonewille paper not present in their mice? Can the authors really rule out the simple explanation that navigation impairments in the L7-PP2B mice may have a vestibular origin? The impact of the paper would be much reduced if the navigation impairments of L7-PP2B mice could be ascribed to vestibular/motor problems, given the well-established role of the cerebellum in motorfunction.

We wish to here report exactly the motor/vestibular deficit described in the paper of Schonewille et al., (Neuron, 2010). The authors described that L7-PP2B mice have no basic motor deficit affecting locomotion (no ataxia as illustrated by openfield and footprint analysis, Fig S3), no deficit in the kinetics of their eyelid responses but are “moderately affected in the performance of their basic compensatory eye movements”. This moderate deficit is found in both the optokinetic reflex and vestibulo-ocular reflex and seems to reflect more a general moderate deficit in image stabilization on the retina than a problem in vestibular reflexes. Furthermore, spared performances of L7-PP2B mice during visually guided navigation (Supplementary Fig. 10 and the light part of Supplementary Fig S12 of the present manuscript) demonstrate their ability to use visual cues to navigate despite this moderate deficit in image stabilization.

We dismiss reviewer’s concerns because we have performed a battery of sensori-motor tasks assessing anxiety, equilibrium, locomotion and motor coordination, and found no deficit except for one of the two constant speeds tested in the rotarod (Supplementary Fig. 1). We also investigated exploratory behavior of L7-PP2B and found no differences with control mice (Supplementary Fig 2). We have then further analyzed basic sensori-motor abilities and exploratory behavior in the dark, during which vestibular information is primarily used and found here again that all the parameters were similar between control and L7-PP2B mice (Supplementary Fig 9). This does not call into questions the deficit described in the Schonewille paper which is essentially a “moderate deficit in compensatory eye movement” (see above). If navigation impairment of L7-PP2B mice had a vestibular origin, we should obtain similar performances to the one observed in passively disoriented mice which is not the case (Supplementary Fig 11). Finally, the nature of the deficits observed in L7-PP2B place cells does not fit with the phenotype observed after inactivation of the vestibular system which led to the disruption of location-specific firing in hippocampal place cells (Stackman et al., 2002, Russel et al., 2003).

Finally, to decipher if spatial performances that were altered in the spatial version of the Morris Water Maze might be explained by a difference in swimming abilities, we also 1) analyzed if there was a correlation between altered performances and speed at the individual level, and found no correlation (Supplementary Fig 10) and 2) specifically compared control and L7-PP2B mice performances during the first session of training (before any learning occurs) and found here again no statistical differences between the two groups (Fig. 6).

Given the factual absence of deficit found in the diverse experiments we performed, we thus dismiss the concern raised by the reviewer of a basic motor/vestibular problem that could explain the deficit

in spatial representation and/or the deficit in spatial navigation observed in these mice. In addition, the fact that L7-PP2B mice behave differently than mice that have been externally disoriented (and thus mice whose vestibular system has been disturbed) confirm that the deficit observed in L7-PP2B mice described in this paper cannot be simply explained by a deficit in vestibular reflexes.

Overall Evaluation:

The original paper suggested an interesting dichotomy between the roles of cerebellar LTP in allocentric-based navigation (examined in this paper using L7-PP2B mice), and cerebellar LTD in self motion-based navigation (examined using L7-PKC mice in a previous paper by the same authors). Although this dichotomy continues to be emphasized in the most recent version of the manuscript, new experiments and analyses have demonstrated that the contribution of cerebellar LTD and LTP to spatial navigation cannot be dissociated. There does seem to be an important difference in the way that cerebellar LTD and LTP contribute to spatial representations in place cells of the hippocampus. In their previous paper, the authors used L7-PKC mice to show that Purkinje LTD is required for maintaining place cell stability that relies on self motion cues in the dark, but not for maintaining place cell stability that relies on allocentric visual cues in the light. In the current paper, the authors use the L7-PP2B mouse to show that Purkinje LTP is required for maintaining place cell stability in the light but not in the dark. This difference in place cell stability between L7-PKC and L7-PP2B mice is the most notable result of the paper, but its impact is reduced for the following reasons:

1) There is nothing in the paper that speaks to the behavioral consequences of the difference in place cell stability between the L7-PKC and the L7-PP2B mice. Indeed, the authors admit in their last rebuttal that there is not a causal link between place cell stability of the L7-PP2B mice and their performance during spatial navigation.

2) Previous work has reported that L7-PP2B mice have vestibular/motor problems (Schonewille et al., Neuron). Thus, the possibility that a simple problem in vestibular/motor function could be the origin of the observed impairments in spatial navigation in the L7-PP2B cannot be ruled out. Given the data presented, there is nothing to support the claim that impaired Purkinje LTP in the L7-PP2B mice leads to deficits in spatial navigation via effects on the hippocampus. If the spatial navigation deficits of the L7-PP2B were caused by vestibular/motor dysfunction (and not by hippocampus problems), the impact of such a finding would be reduced because the link between impaired cerebellar plasticity and motor dysfunction is well-known.

The fact that, at the behavioral level, allocentric and self-motion based navigation are both altered without cerebellar LTD or LTP do not invalidate the dichotomy found at the level of the hippocampal spatial representation, which has always been the centerpiece of our manuscript.

In addition, we have always presented a direct behavioral correlate of place cell instability in the presence of an increased object exploration that occurred specifically during spatial map disorientation (Figure 2).

Finally, as explained above we have several factual data to rule out the possibility that a simple problem in vestibular/motor function could be the origin of the observed impairments in spatial navigation. The aim of our study was to evaluate how a lack of cerebellar potentiation might affect spatial representation in the hippocampus on one hand and goal-directed navigation at the behavioral level on the other hand. We presented direct evidences supporting the claim that impaired cerebellar Purkinje potentiation leads to an unstable hippocampal spatial coding and navigation deficits.

Reviewers' comments:

Reviewer #1 (Remarks to the Author):

The paper has been further improved by the addition of new experiments and I think it presents a novel set of work that is interesting and moves the field forward. It is true that a complete mechanistic understanding of the observed place cell effect is still lacking but I feel that this goes beyond the scope of the current manuscript. I do have a few minor comments.

The paper needs to be edited/checked by a copy editor – there are a significant number of grammatical errors that need to be corrected.

Figure 2B – state what the N and n refer to in the figure legend.

Figure 2 C-G; I'm not sure an $n = 3$ (control unstable) is a large enough sample size to run statistics. At the very least – the bar graphs should be removed – as a sample size that small should only be presented as individual data points (with mean values).

Reviewer #1 (Remarks to the Author):

The paper has been further improved by the addition of new experiments and I think it presents a novel set of work that is interesting and moves the field forward. It is true that a complete mechanistic understanding of the observed place cell effect is still lacking but I feel that this goes beyond the scope of the current manuscript. I do have a few minor comments.

The paper needs to be edited/checked by a copy editor – there are a significant number of grammatical errors that need to be corrected.

Figure 2B – state what the N and n refer to in the figure legend.

Figure 2 C-G; I'm not sure an $n = 3$ (control unstable) is a large enough sample size to run statistics. At the very least – the bar graphs should be removed – as a sample size that small should only be presented as individual data points (with mean values).

We thank the reviewer for his/her helpful comments throughout the different revisions of this manuscript. We have now indicated the meaning of the N and n in the legend of the figure 2. We also agree that $n=3$ is a too small sample size to run statistics. We thus removed the bar graphs representing the mean and only present the three individual data points according to reviewer's suggestion (see new Fig.2)